# Spectral Regularization Allows Data-frugal Learning over Combinatorial Spaces

**Amirali Aghazadeh**                                    *amiralia@gatech.edu*
*School of Electrical and Computer Engineering*
*Georgia Institute of Technology*

**Nived Rajaraman**                              *nived.rajaraman@berkeley.edu*
*Department of Electrical Engineering & Computer Science*
*University of California, Berkeley*

**Tony Tu**                                               *ttu32@gatech.edu*
*School of Computer Science*
*Georgia Institute of Technology*

**Kennan Ramchandran**                           *kannanr@eecs.berkeley.edu*
*Department of Electrical Engineering & Computer Sciences*
*University of California, Berkeley*

**Reviewed on OpenReview:** *https://openreview.net/forum?id=mySiFHCeAl&noteId=WhfpRCk8Wz*

## Abstract

Data-driven machine learning models are being increasingly employed in several important inference problems in biology, chemistry, and physics, which require learning over combinatorial spaces. Recent empirical evidence (see, e.g., Tseng et al. (2020); Aghazadeh et al. (2021); Ha et al. (2021)) suggests that regularizing the spectral representation of such models improves their generalization power when labeled data is scarce. However, despite these empirical studies, the theoretical underpinning of when and how spectral regularization enables improved generalization is poorly understood. In this paper, we focus on learning pseudo-Boolean functions and demonstrate that regularizing the empirical mean squared error by the $L_1$ norm of the spectral transform of the learned function reshapes the loss landscape and allows for data-frugal learning under a restricted secant condition on the learner's empirical error measured against the ground truth function. Under a weaker quadratic growth condition, we show that stationary points, which also approximately interpolate the training data points achieve statistically optimal generalization performance. Complementing our theory, we empirically demonstrate that running gradient descent on the regularized loss results in a better generalization performance compared to baseline algorithms in several data-scarce real-world problems.

## 1 Introduction

Machine learning (ML) models have become increasingly commonplace in learning pseudo-Boolean functions $f(\cdot)$, which map a $d$-dimensional binary vector $x \in \{\pm 1\}^d$ to a real number $f(x) \in \mathbb{R}$. In biology, ML models are used to infer the functional properties of macro-molecules (e.g., proteins) from a small set of mutations (Riesselman et al., 2018; Gelman et al., 2021). In physics, ML models are being used to infer the thermodynamic properties of combinatorial systems defined over a set of binary (Ising) state variables (Carleo et al., 2019; Noé et al., 2019). These highly nonlinear and complex ML models are trained using modern techniques in continuous optimization. Yet they inherit the elegant properties of pseudo-Boolean functions over the discrete binary hypercube $\{\pm 1\}^d$. In particular, it is known that the spectral representation of

a pseudo-Boolean function is defined as the Walsh-Hadamard transform (WHT) of the resulting vector of combinatorial function evaluations, that is, $\mathbf{H}f(\mathbf{X})$, where $\mathbf{H}$ is a $2^d \times 2^d$ Walsh matrix and the function evaluation vector $f(\mathbf{X}) = [f(x) : x \in \mathbf{X}]^T$ is sorted in the lexicographic ordering of all the length-$d$ binary strings in $\mathbf{X}$. The WHT enables writing the pseudo-Boolean function $f(x)$ as a multi-linear polynomial $f(x) = \sum_{\mathcal{S} \subseteq [d]} \alpha_{\mathcal{S}} \prod_{i \in \mathcal{S}} x_i$, where $[d] = \{1, 2, \ldots, d\}$ and $\alpha_{\mathcal{S}} \in \mathbb{R}$ is the WHT coefficient corresponding to the monomial $\prod_{i \in \mathcal{S}} x_i$ (Boros & Hammer, 2002).[1]

Recent studies in biology (e.g., protein function prediction) have measured the output of several of these real-world functions $f(x)$ to all the $2^d$ enumerations of the input $x$ and analyzed their spectral representation. These costly high-throughout experiments on combinatorially complete datasets demonstrate a curious phenomenon: such pseudo-Boolean functions often have low dimensional structures that manifest in the form of an approximately-sparse polynomial representation (i.e., approximately-sparse WHT) with the top-$k$ coefficients corresponding to physically-meaningful interactions (Poelwijk et al., 2019; Eble et al., 2020; Brookes et al., 2022) (see Appendix A for empirical evidence on protein functions).

The problem of learning pseudo-Boolean functions with sparse polynomial representations has a rich history from theoretical and empirical viewpoints. In particular, since the pseudo-Boolean functions being learned in their polynomial representations are linear functions in their coefficients, one may think about this problem as a special instance of *sparse linear regression* in dimension $2^d$. There are many algorithms for sparse linear regression, such as LARS (Efron et al., 2004), OMP (Tropp & Gilbert, 2007), AMP (Donoho et al., 2009), and FISTA (Beck & Teboulle, 2009). In particular, one may apply LASSO to get statistical guarantees for this problem which only scale linearly in the sparsity of the underlying ground truth polynomial $k$ and logarithmically in the problem dimension $\log(2^d)$ (Candes et al., 2006). Likewise, from another perspective, one may view the polynomial instead as a vector of $2^d$ outcomes $f(\mathbf{X})$; the objective of the learner is to approximate $f(\mathbf{X})$, when the learner can observe any chosen set of a few entries of this vector (corrupted by noise), under the assumption that $f(\mathbf{X})$ itself has a sparse WHT. In the literature, this problem has been referred to by several names, and we refer to it as the *sparse WHT problem*. There are many computationally and statistically efficient algorithms for solving this problem, such as SPRIGHT (Li & Ramchandran, 2015) and Sparse-FFT (Amrollahi et al., 2019), among others.

A common issue with these alternate views of the problem, such as sparse linear regression or sparse WHT is that the resulting algorithms are not suited for *function approximation*. In particular, real-world physical and biological functions often have additional structures which are either unknown *a priori* or cannot be described succinctly, and which nonlinear function classes are often implicitly biased towards learning. Indeed, several deep neural networks (DNNs) have been shown to exhibit strong generalization performance in biological and physical problems (Gelman et al., 2021; Ching et al., 2018; Sarkisyan et al., 2016). This leads to a fundamental disconnect between algorithms for which strong theoretical guarantees are known (e.g., LASSO) and practical ML models based on nonlinear and complex function classes which are well-suited for function approximation.

Another issue is specific to algorithms for solving the sparse WHT problem: some of these approaches require observing $f(\cdot)$ at any chosen input (Li & Ramchandran, 2015; Li et al., 2015). In many practical applications, where data is prohibitively expensive to collect, one is often forced to work with a handful of random samples. For example, in the case of proteins, a common approach to measure biological functions is through a procedure called *random mutagenesis*, which allows only a random sampling of the combinatorial sequence space Sarkisyan et al. (2016). This constraint renders several algorithms for the sparse WHT problem ill-suited for learning, even though they admit near-optimal computational and sample complexity guarantees.

From a more practical point of view, several recent approaches for learning over combinatorial spaces (see, e.g., (Tseng et al., 2020; Aghazadeh et al., 2021; Ha et al., 2021; Li et al., 2020)) follow similar ideas for improving empirical generalization. Instead of directly learning the $2^d$-dimensional polynomial, they have converged on *explicitly* promoting sparsity in the spectral representation of the learned compactly-represented nonlinear function—also known as spectral regularization. These empirical studies motivate several important theoretical questions regarding the underlying mechanism of spectral regularization in

---

[1]We use these terms interchangeably for pseudo-Boolean functions: spectral, Fourier, and Walsh-Hadamard.

improving generalization in practice. This paper focuses on the question: **When and how does spectral regularization improve generalization?** The answer to this question is important from theoretical and practical perspectives. Theoretically, it helps us understand how the loss landscape is reshaped as the result of spectral regularization in favor of data-frugal learning. Practically, it tells us when and how to use spectral regularization in learning real-world combinatorial functions.

**Contributions**. In this paper, we theoretically analyze spectral regularization of the empirical risk from the perspective of learning pseudo-Boolean functions. We demonstrate that regularizing with the $L_1$ norm of the spectrum of the learned function improves the generalization performance of the learner. In particular, under a particular Restricted Secant Inequality (RSI) condition on the learner's empirical error, measured against the ground truth, stationary points of the regularized loss admit optimal generalization performance. We relax this to a weaker quadratic growth (QG) condition and show that, under an additional data-interpolation condition, stationary points of the regularized loss also admit statistical optimality. We empirically demonstrate that these conditions are satisfied for a wide class of nonlinear functions. Our analysis provides a new generalization bound when the underlying learned functions are sparse in the spectral domain. Empirical demonstrations on several real-world data sets complement our theoretical findings.

## 2 Learning Pseudo-Boolean Functions

**Problem statement**. We consider the problem of learning structured pseudo-Boolean functions on the binary hypercube $\{\pm 1\}^d$. In the finite sample setting, we assume that the learner has access to a dataset $D_n$ comprised of $n$ labeled data points $(x^i, y^i)_{i=1}^n$ where $x^i \in \{\pm 1\}^d$ are drawn from an input distribution $\mathcal{D}$ and the real-valued output $y_i \in \mathbb{R}$ is a noisy realization of an *unknown* pseudo-Boolean function of the input [2]. Consider a rich nonlinear function class $\mathcal{F} = \{f_{\boldsymbol{\theta}} : \boldsymbol{\theta} \in \mathbb{R}^m\}$ parameterized by $\boldsymbol{\theta}$, where $f_{\boldsymbol{\theta}} : \{\pm 1\}^d \to \mathbb{R}$. We study the realizable setting, where the ground-truth labels are generated as $y^i = f_{\boldsymbol{\theta}^*}(x^i) + Z^i$ for an unknown parameter $\boldsymbol{\theta}^* \in \mathbb{R}^m$, where $Z^i$ is the noise in the measurement for input $x^i$, assumed to be independent and normally distributed $\sim \mathcal{N}(0, \sigma^2)$ [3]. The objective of the learner is to learn $\boldsymbol{\theta}^*$.

*Multi-linear polynomial representation.* Any pseudo-Boolean function $f(\cdot)$ can be uniquely represented as a multi-linear polynomial, $f(x) = \sum_{z \in \{\pm 1\}^d} \alpha_z \prod_{i:z_i=+1} x_i$, where the scalar $\alpha_z \in \mathbb{R}$ is the coefficient corresponding to the monomial $\prod_{i:z_i=+1} x_i$, with order $\sum_{i=1}^d \mathbb{1}(z_i = +1)$ (Boros & Hammer, 2002). The problem of learning a pseudo-Boolean function $f(\cdot)$ is equivalent to finding the $2^d$ unknown coefficients $\alpha_z$. In particular, the evaluations of $f$ on all vertices of the binary hypercube $\{\pm 1\}^{2^d}$ results in a linear system of equations over the $\alpha_z$ variables,

$$\left[f(x) : x \in \{\pm 1\}^d\right]^T = \mathbf{H} \left[\alpha_z : z \in \{\pm 1\}^d\right]^T, \tag{1}$$

where the variables $x$ and $z$ enumerate the vertices of the binary hypercube $\{\pm 1\}^d$ in lexicographic order, and $\mathbf{H} \equiv \mathbf{H}_{2^d}$ denotes the $2^d \times 2^d$ scaled Hadamard matrix defined recursively as,

$$\mathbf{H}_{2^{d+1}} = \frac{1}{\sqrt{2}} \begin{bmatrix} \mathbf{H}_{2^d} & \mathbf{H}_{2^d} \\ \mathbf{H}_{2^d} & -\mathbf{H}_{2^d} \end{bmatrix}, \tag{2}$$

where $\mathbf{H}_1 = \begin{bmatrix} +1 \end{bmatrix}$. Thus the vector of function evaluations can be obtained by taking the WHT of the vector of coefficients in the polynomial representation of $f$. Furthermore, by inverting the above linear system (note that $\mathbf{H}$ is an orthonormal matrix), the vector of polynomial coefficients $[\alpha_z : z \in \{\pm 1\}^d]$ can, in turn, be obtained by taking the WHT of the vector $\left[f(x) : x \in \{\pm 1\}^d\right]^T$. For brevity of notation, for a function $f$, we define $f(\mathbf{X})$ as $\left[f(x) : x \in \{\pm 1\}^d\right]^T$ where in the LHS the binary strings are enumerated in lexicographic order. The coefficients in the polynomial representation of $f(\cdot)$ can be collected in the vector $\mathbf{H}f(\mathbf{X})$. For functions, we use "sparsity in WHT" and "sparse polynomial representation" interchangeably.

---

[2] We use the subscript $x_i$ to refer to the $i^{\text{th}}$ digit in the binary string $x = (x_1, x_2, \cdots, x_d)$.

[3] In fact, our results only require the noise to be independent subgaussian random variables with variance parameter $\sigma^2$, but for ease of exposition, we impose the Gaussian noise condition.

*Sparsity in real-world functions.* Even in the noiseless setting, to perfectly learn a general pseudo-Boolean function $f(\cdot)$, its evaluations on all binary input vectors $x \in \{\pm 1\}^d$ are required, which may be very expensive. In the presence of additional structures, this significant statistical and computational cost can be mitigated. One such structure that has been observed in real-world biological and physical functions is sparsity in WHT (Poelwijk et al., 2019; Eble et al., 2020; Brookes et al., 2022). In particular, in the theoretical analysis of this paper, we assume that the ground-truth function $f_{\boldsymbol{\theta}^*}$ has a *sparse polynomial representation* composed of at most $k$ monomials. Namely, $\|\mathbf{H}f_{\boldsymbol{\theta}^*}(\mathbf{X})\|_0 \leq k$, where $k \ll 2^d$ is typically unknown.

**Spectral regularization**. In attempting to learn pseudo-Boolean functions with a sparse polynomial representation, a natural question to ask is: how can one encourage the learned functions also to have sparse polynomial representations. One solution is to regularize the training loss with an additional functional which promotes sparsity in the polynomial representation of the learned function $\widehat{f}$. Denoting the polynomial representation of $\widehat{f}$ by $\sum_{z \in \{\pm 1\}^d} \widehat{\alpha}_z \prod_{i:z_i=+1} x_i$, a natural regularization functional would be $\|\widehat{\boldsymbol{\alpha}}\|_0 \equiv \|\mathbf{H}\widehat{f}(\mathbf{X})\|_0$ which is also the sparsity of $f$ in WHT.

However, since $\| \cdot \|_0$ is not a continuous function, we proxy the $\| \cdot \|_0$ by the $L_1$ norm. The resulting regularization functional is, $\|\mathbf{H}\widehat{f}(\mathbf{X})\|_1$. When learning over a parameterized function family, $\widehat{f} = f_{\boldsymbol{\theta}}$, the resulting regularized ERM we consider in this paper is,

$$\min_{\boldsymbol{\theta}} \mathcal{L}_n(\boldsymbol{\theta}) + R(\boldsymbol{\theta}), \text{ where } R(\boldsymbol{\theta}) \triangleq \frac{\lambda}{\sqrt{2^d}}\|\mathbf{H}f_{\boldsymbol{\theta}}(\mathbf{X})\|_1, \tag{OBJ}$$

where $\mathcal{L}_n(\boldsymbol{\theta}) \triangleq \frac{1}{n} \sum_{i=1}^n \left[ \left( f_{\boldsymbol{\theta}}(x^i) - y^i \right)^2 \right]$ is the empirical mean squared error (MSE). Regularizing by $R(\boldsymbol{\theta})$ of the above form is known as *spectral regularization (SP)*.

**Remark 1.** *The scaling of the regularization parameter, $\frac{\lambda}{\sqrt{2^d}}$, is motivated from the linearly parameterized setting of $\mathcal{F} = \{\langle \boldsymbol{\theta}, x \rangle : \boldsymbol{\theta} \in \mathbb{R}^d\}$. An explicit computation gives, $\frac{\lambda}{\sqrt{2^d}}\|\mathbf{H}f_{\boldsymbol{\theta}}(\mathbf{X})\|_1 = \lambda\|\boldsymbol{\theta}\|_1$, which is the scaling of the regularization parameter as used in LASSO.*

The regularization weight $\lambda > 0$ strikes a balance between the empirical MSE and the spectral regularization, and is set empirically using cross validation. Since the aggregate loss $\mathcal{L}_n(\boldsymbol{\theta}) + R(\boldsymbol{\theta})$ is subdifferentiable with respect to the model parameters $\boldsymbol{\theta}$, we can apply stochastic (sub-)gradient methods on the aggregate loss. Note however that, in general, as a function of the parameter $\boldsymbol{\theta}$, both the MSE, as well as the SP regularization are non-convex functions.

## 3  Related Works

A recent line of theoretical works on learning pseudo-Boolean functions demonstrate a staircase-like property of gradient descent in learning DNNs in that the WHT coefficients corresponding to higher order monomials (e.g., $x_1 x_2 x_3$) are reachable from lower order ones along increasing chains (i.e., $x_1 x_2$ and $x_1$), and are thus learnable in polynomial time and sample cost (Abbe et al., 2021). Other works have shown that under certain distributions, low order parity functions are learnable by means of gradient decent on depth-2 networks, while they cannot be learned efficiently using linear methods (Daniely & Malach, 2020). These analyses are limited to certain DNN architectures or assume (linear) approximations to DNNs (e.g., neural tangent kernels) which entirely disallows the analysis of spectral regularization as they manifest only in nonlinear function classes.

Spectral bias of DNNs have been the subject of several other empirical and theoretical works (Yang & Salman, 2019). Approximations to the Fourier transform of two-layer (Zhang et al., 2019) and multi-layer (Rahaman et al., 2019) ReLU networks show that these networks have a learning bias towards low frequency functions (Xu et al., 2019). To improve the limitations of DNNs in learning high frequency components, empirical works use Fourier features explicitly as part of the input (Tancik et al., 2020). A parametrization of polynomial DNNs has also been shown to speed up the learning of higher frequency components in two-layer networks (Choraria et al., 2022). Different notions of spectral priors have also been empirically investigated in graph neural networks (Li et al., 2020) and elsewhere (Yoshida & Miyato, 2017). These results support the implicit bias of DNNs towards dense and low-frequency spectral representation and only motivate our analysis of sparsity as an explicit spectral prior.

## 4 Theoretical Analysis

To develop some intuition about the general problem, consider the special case of learning over the set of linear functions. Namely, $\mathcal{F} = \{\langle \boldsymbol{\theta}, \cdot \rangle : \boldsymbol{\theta} \in \mathbb{R}^d\}$. The polynomial representation of any linear function $f_{\boldsymbol{\theta}}(x) = \langle \boldsymbol{\theta}, x \rangle$ is simply $\sum_{i \in [d]} \theta_i x_i$ where only the order-1 coefficients are non-zero. Thus, the assumption that the polynomial representation of $f_{\boldsymbol{\theta}}$ is composed of at most $k$ monomials is equivalent to saying that $\boldsymbol{\theta}$ is $k$-sparse. Thus in the linear setting, the problem boils down to sparse linear regression (Candes et al., 2006). Furthermore, in the linear setting, spectral regularization has an explicit representation in terms of its parameter $\boldsymbol{\theta}$. In particular, here $R(\boldsymbol{\theta}) = \frac{\lambda}{\sqrt{2^d}} \|\mathbf{H} f_{\boldsymbol{\theta}}(\mathbf{X})\|_1 = \lambda \|\boldsymbol{\theta}\|_1$. Thus, the proposed objective function, (OBJ), when specialized to the linear setting boils down to the LASSO objective, $\frac{1}{n} \sum_{i=1}^n (\langle x^i, \boldsymbol{\theta} \rangle - y^i)^2 + \lambda \|\boldsymbol{\theta}\|_1$, which can be efficiently solved by gradient descent, and is known to be statistically optimal in the finite sample regime (Raskutti et al., 2011).

### 4.1 Going beyond the linear setting - Restricted Secant Inequality

In the linear setting, a sufficient condition for the finite sample statistical optimality of LASSO is the restricted eigenvalue condition on the empirical covariance matrix $\Sigma_n = \frac{1}{n} \sum_{i=1}^n x^i x^{i^T}$. The restricted eigenvalue condition is automatically satisfied if the smallest eigenvalue of $\Sigma_n$ is bounded away from 0. The condition induces a particular sense of curvature around the true parameter $\boldsymbol{\theta}^*$ of the loss $\mathrm{Err}_n(\boldsymbol{\theta}, \boldsymbol{\theta}^*)$, that is the MSE of the learned function compared to the ground truth,

$$\mathrm{Err}_n(\boldsymbol{\theta}, \boldsymbol{\theta}^*) = \frac{1}{n} \sum_{i=1}^n (f_{\boldsymbol{\theta}}(x^i) - f_{\boldsymbol{\theta}^*}(x^i))^2. \tag{3}$$

In contrast, when $\mathcal{F}$ is a non-linear function family, we circumvent these sufficient conditions, and directly study assumptions which induce curvature in the loss $\mathrm{Err}_n(\boldsymbol{\theta}, \boldsymbol{\theta}^*)$. These assumptions, however, do not concern the structure of the spectral regularizer, which unlike in the linear case, can no longer be represented as a closed-form function of $\boldsymbol{\theta}$. A major technical challenge in the analysis relates to circumventing the use of this explicit representation, which we expand upon later.

**Definition 1** (Restricted secant inequality (RSI) (Zhang & Yin, 2013)). *The set of functions satisfying the restricted secant inequality with parameter $C$, denoted* $\mathrm{RSI}(C)$ *is defined as follows. A function* $g : \mathbb{R}^m \to \mathbb{R}$ *belongs to* $\mathrm{RSI}(C)$ *iff for some* $z^* \in \arg\min_{z \in \mathbb{R}^m} g(z)$ *and for all* $z \in \mathbb{R}^m$,

$$\langle \nabla g(z), z - z^* \rangle \geq C \|z - z^*\|_2^2. \tag{4}$$

In other words, the restricted secant inequality implies that the gradient of the loss at a point is well correlated with the $z - z^*$, the line joining the current point to the minimizer of $g$.

**Remark 2.** *The RSI is known to generalize several extensions of convexity, such as quasar-star convexity (Hinder et al., 2020) and strong star-convexity (Lee & Valiant, 2015). Refer to Karimi et al. (2016) for a comparison with other constraints such as the Polyak-Lojasiewicz (PL) inequality, and the quadratic growth condition which we study in Section 4.2. In general, the RSI is a significantly weaker condition than global strong convexity.*

The first main assumption we study is when the $\mathrm{Err}_n(\boldsymbol{\theta}, \boldsymbol{\theta}^*)$ satisfies the RSI. This assumption captures a notion of curvature, in which the gradients of $\mathrm{Err}_n(\boldsymbol{\theta}, \boldsymbol{\theta}^*)$ are informative about (i.e., positively correlated with) the error in the parameter space $\boldsymbol{\theta} - \boldsymbol{\theta}^*$.

From this behavior, it may be expected that all stationary points of functions which satisfy the RSI are global minima, which is indeed true (Karimi et al., 2016). However, note that we impose the RSI condition on $\mathrm{Err}_n(\boldsymbol{\theta}, \boldsymbol{\theta}^*)$ (which cannot be computed by the learner) and not the empirical risk $\mathcal{L}_n(\boldsymbol{\theta})$. Even under the RSI condition on $\mathrm{Err}_n(\boldsymbol{\theta}, \boldsymbol{\theta}^*)$, it is not trivial to find global minima of the training objective, $\mathcal{L}_n(\boldsymbol{\theta}) + R(\boldsymbol{\theta})$ which is composed of two non-convex (in $\boldsymbol{\theta}$) functions and is non-smooth. Thus, we restrict our attention to the analysis of *first-order stationary points* of this objective.

**Assumption 1(a).** *Assume the function* $\mathrm{Err}_n(\boldsymbol{\theta}, \boldsymbol{\theta}^*)$ *satisfies the RSI with high probability over the dataset. Namely, there is a constant* $C_{n,\delta}^* > 0$ *such that with probability at least* $1 - \delta$, *for every* $\boldsymbol{\theta} \in \mathbb{R}^m$,

$$\langle \boldsymbol{\theta} - \boldsymbol{\theta}^*, \nabla_\theta \mathrm{Err}_n(\boldsymbol{\theta}, \boldsymbol{\theta}^*) \rangle \geq C_{n,\delta}^* \|\boldsymbol{\theta} - \boldsymbol{\theta}^*\|_2^2. \tag{5}$$

**Remark 3.** *The RSI is true for linear families* $\mathcal{F} = \{\langle \boldsymbol{\theta}, \cdot \rangle : \boldsymbol{\theta} \in \mathbb{R}^m\}$ *as long as with probability at least* $1 - \delta$, *the data covariance satisfies* $\mathbb{E}_{x \sim \mathrm{Unif}(D_n)}[xx^T] \succeq C_{n,\delta}^* I$, *i.e., is well conditioned. The quadratic growth condition in Section 4.2 also holds under the same conditions with parameter* $C_{n,\delta}^*$.

In addition to the RSI, we make some mild regularity assumptions on the function classes we study. We emphasize that none of these assumptions imply convexity or smoothness of the training objective $\mathcal{L}_n(\boldsymbol{\theta}) + R(\boldsymbol{\theta})$ or $\mathrm{Err}_n(\boldsymbol{\theta}, \boldsymbol{\theta}^*)$. First, we assume that the function class $\mathcal{F}$ is sufficiently smooth, having Lipschitz continuous gradients (Cisse et al., 2017). Note that we study the case when the function class $\mathcal{F}$ is smooth, and do not impose this condition on the empirical risk, $\mathcal{L}_n(\boldsymbol{\theta})$, as is often assumed (Jin et al., 2018).

**Assumption 2** (Lipschitz continuous gradients on $\mathcal{F}$). *For each* $x \in \mathbf{X}$ *and constant* $\mu$, *assume that* $f_{\boldsymbol{\theta}}(x)$ *is twice differentiable in* $\boldsymbol{\theta}$ *and the Hessian of* $f_{\boldsymbol{\theta}}(x)$ *satisfies* $-\mu I \preceq \nabla^2 f_{\boldsymbol{\theta}}(x) \preceq \mu I$.

The final assumption we impose assumes that the ground truth function $\boldsymbol{\theta}^*$ has well behaved gradients in that a certain covariance matrix induced by the gradients of the matrix is bounded.

**Assumption 3** (Bounded average gradient norm at $\boldsymbol{\theta}^*$). *Under this assumption,*

$$\mathbb{E}_{x \sim \mathrm{Unif}(\mathbf{X})} \left[ \nabla f_{\boldsymbol{\theta}^*}(x)(\nabla f_{\boldsymbol{\theta}^*}(x))^T \right] \preceq L^2 I. \tag{6}$$

Assumption 3 can be interpreted as a one-point Lipschitzness condition, showing that on average across the inputs $x \in \mathbf{X}$, the gradients of $f_{\boldsymbol{\theta}}(x)$ are well behaved at the singular point $\boldsymbol{\theta} = \boldsymbol{\theta}^*$. Under the previous assumptions, we prove a bound on the error in the parameter space made by the learner. This theorem is a consequence of a more general result we prove in Appendix C for an arbitrary regularization scale $\lambda > 0$.

**Theorem 1.** *Suppose Assumptions 1(a), 2 and 3 are satisfied,* $\lambda$ *is chosen as* $12\sigma\sqrt{\frac{d + \log(1/\delta)}{n}}$ *and the size of the dataset is sufficiently large, namely* $n > n_0$, *defined as,*

$$n_0 \triangleq C_1 \frac{\sigma^2 \left(d^2\mu^2 + L^2k\right)(d + \log(1/\delta))}{(C_{n,\delta}^*)^2}, \tag{7}$$

*for a large constant* $C_1 < 1000^4$. *Consider a learner which returns a first order stationary point,* $\hat{\boldsymbol{\theta}}$, *of the loss* $\mathcal{L}_n(\boldsymbol{\theta}) + R(\boldsymbol{\theta})$. *Then, with probability* $\geq 1 - 2\delta$,

$$\|\hat{\boldsymbol{\theta}} - \boldsymbol{\theta}^*\|_2 \lesssim \frac{\sigma L}{C_{n,\delta}^*} \sqrt{\frac{k(d + \log(1/\delta))}{n}}. \tag{8}$$

*Note that in the dependence on* $d$ *and* $k$, *the sample size threshold* $n_0$ *scales asymptotically as* $d^3 + dk$.

## 4.2 Going beyond the RSI - The Quadratic Growth condition

While RSI is a much weaker condition than convexity, at a high level, it assumes that the gradients of $\mathrm{Err}_n(\boldsymbol{\theta}, \boldsymbol{\theta}^*)$ are informative about the error in the parameter space $\boldsymbol{\theta} - \boldsymbol{\theta}^*$. We can further relax this assumption. In this section, we consider a weakening of this assumption which only supposes that $\mathrm{Err}_n(\boldsymbol{\theta}, \boldsymbol{\theta}^*)$ grows at least quadratically in $\|\boldsymbol{\theta} - \boldsymbol{\theta}^*\|_2$. This is known as the *quadratic growth* (QG) condition (Anitescu, 2000). This condition no longer characterizes the behavior of the gradients of the function $\mathrm{Err}_n(\boldsymbol{\theta}, \boldsymbol{\theta}^*)$, much less those of the empirical loss $\mathcal{L}_n$. Stationary points of a function satisfying the QG condition are no longer global minima, in contrast with the behavior under the RSI (Definition 1).

---

[4]Note that we did not optimize the value of this constant. With a more careful analysis, its value can be brought down.

**Definition 2** (Quadratic Growth (QG) condition (Anitescu, 2000)). *The set of functions satisfying the QG condition with parameter $C$, denoted $\mathrm{QG}(C)$ is defined by the inclusion, $g : \mathbb{R}^m \to \mathbb{R}$ belongs to $\mathrm{QG}(C)$ iff for some $z^* \in \arg\min_{z \in \mathbb{R}^m} g(z)$ and for all $z \in \mathbb{R}^m$,*

$$g(z) - g(z^*) \geq C\|z^* - z\|_2^2. \tag{9}$$

*For $C > 0$, functions in $\mathrm{QG}(C)$ have a unique global minimizer.*

As the name suggests, the quadratic growth condition implies that the function $g$ has local curvature around $z^*$. However, it is important to note that this does not discount the possibility that $g$ has many spurious local minima. In this section, we go beyond the RSI assumption on $\mathrm{Err}_n(\boldsymbol{\theta}, \boldsymbol{\theta}^*)$ and discuss the case where it satisfies the quadratic growth condition.

**Assumption 1(b)** (Quadratic Growth (QG) condition (Anitescu, 2000)). *The function family $\mathcal{F} = \{f_{\boldsymbol{\theta}} : \boldsymbol{\theta} \in \mathbb{R}^m\}$ is assumed to satisfy the QG condition with parameter $C^*_{n,\delta}$, if with probability at least $1 - \delta$ over the dataset $D_n$,*

$$\forall \boldsymbol{\theta} \in \mathbb{R}^m, \ \mathrm{Err}_n(\boldsymbol{\theta}, \boldsymbol{\theta}^*) \geq C^*_{n,\delta}\|\boldsymbol{\theta} - \boldsymbol{\theta}^*\|_2^2. \tag{10}$$

**Remark 4.** *(Anitescu, 2000, Theorem 2) Restricting to functions $f$ which have $L$-Lipschitz continuous gradients, if $f \in \mathrm{RSI}(C)$, then it implies that $f \in \mathrm{QG}(2C/L)$. This implies that up to the value of the parameter, the QG condition is weaker than RSI, under a smoothness constraint on the considered functions.*

Note that under the QG condition, gradients of $\mathrm{Err}_n(\boldsymbol{\theta}, \boldsymbol{\theta}^*)$ are no longer constrained to be positively correlated with the $\boldsymbol{\theta} - \boldsymbol{\theta}^*$. In fact, the absence of this feature proves to be a significant barrier for first-order methods to generalize well. To circumvent this issue, we motivate and introduce the notion of *approximate first-order stationary interpolators* in the next section, and characterize their statistical performance. This imposes a stronger requirement on the algorithm than just returning an arbitrary stationary point of the training objective $\mathcal{L}(\boldsymbol{\theta}) + R(\boldsymbol{\theta})$.

**Approximate First-order Interpolators.** When $\mathcal{F}$ is an expressive family of nonlinear models, it has been observed empirically that standard stochastic gradient methods can be run until the model begins to perfectly interpolate the training data, without hurting test-time performance. This phenomenon has been referred to in the literature as benign overfitting (Bartlett et al., 2020), and has been seen to hold frequently in the training of DNNs in practice, when stochastic gradient descent is run for sufficiently many epochs. In fact, in all our experiments (see Figs. 5 and 6 in Appendix A), we observe that upon running stochastic gradient descent for sufficiently long on the aggregate loss $\mathcal{L}_n(\boldsymbol{\theta}) + R(\boldsymbol{\theta})$, the parameter $\hat{\boldsymbol{\theta}}$ eventually converges to a solution *which interpolates the labelled examples well*, i.e., the mean square error $\mathcal{L}_n(\hat{\boldsymbol{\theta}})$ is upper bounded by a sufficiently small $\Delta > 0$. Note that we *do not* assume that $\Delta$ is so small that the condition essentially imposes that $\hat{\boldsymbol{\theta}}$ is an approximate global minimizer of the training loss. In theory, our results also hold under the stronger condition that $\mathcal{L}_n(\hat{\boldsymbol{\theta}}) + R(\hat{\boldsymbol{\theta}})$ is upper bounded by $\Delta$, which is the loss function being optimized by (stochastic) gradient descent. This phenomenon motivates the following definition of approximate first order stationary interpolators.

**Definition 3** ($\Delta$-approximate first order stationary interpolator). *A point $\hat{\boldsymbol{\theta}}$ is defined to be a $\Delta$-approximate first-order stationary interpolator if,*

1. *$\hat{\boldsymbol{\theta}}$ is a first order stationary point of the aggregate loss $\mathcal{L}_n(\boldsymbol{\theta}) + R(\boldsymbol{\theta})$. Namely,*

$$0 \in (\nabla \mathcal{L}_n)(\hat{\boldsymbol{\theta}}) + (\nabla R)(\hat{\boldsymbol{\theta}}). \tag{11}$$

2. *$f_{\hat{\boldsymbol{\theta}}}$ approximately interpolates the observed training data. Namely, the empirical mean squared error of $\hat{\boldsymbol{\theta}}$ evaluated on the training dataset satisfies,*

$$\mathcal{L}_n(\hat{\boldsymbol{\theta}}) \leq \Delta. \tag{12}$$

*Note that our results hold under the stronger condition $\mathcal{L}_n(\hat{\boldsymbol{\theta}}) + R(\hat{\boldsymbol{\theta}}) \leq \Delta$, the training loss function being minimized by the (stochastic) gradient methods in our experiments.*

We are ready to establish a guarantee on the statistical performance of approximate first-order stationary interpolators.

**Theorem 2.** *Suppose $\lambda$ is chosen $= 12\sigma\sqrt{\frac{d+\log(1/\delta)}{n}}$. Assume that the size of the dataset is sufficiently large, namely, $n > n_0$ (as defined in Equation (7)). Consider a learner returns $\hat{\boldsymbol{\theta}}$ which is a $\Delta$-approximate first order stationary interpolator, where $\Delta \leq C_1(C_{n,\delta}^*/\mu)^2$ for a sufficiently large constant $C_1$. Then, with probability $\geq 1 - 2\delta$,*

$$\|\hat{\boldsymbol{\theta}} - \boldsymbol{\theta}^*\|_2 \lesssim \frac{\sigma L}{C_{n,\delta}^*}\sqrt{\frac{k(d + \log(1/\delta))}{n}}. \tag{13}$$

**Remark 5.** *For pseudo-Boolean functions, the log-covering number in any norm up to log factors is $\approx \log\binom{2^d}{k} \approx kd$. Therefore, an algorithm which returns a function which is an exact minimizer of the mean squared error among all functions with a $k$-sparse polynomial representation, $\boldsymbol{\theta}_k^{\mathrm{LS}}$ admits the guarantee $\|\boldsymbol{\theta}_k^{\mathrm{LS}} - \boldsymbol{\theta}^*\|_2 \lesssim \sqrt{kd/n}$ up to scaling constants, with high probability, up to log factors. However, when $\Delta$ is large, the learner considered in Theorem 2 is not constrained to return the exact minimizer of the squared error (subject to the sparsity constraint). Thus, under the additional condition of first-order stationarity, Theorem 2 imposes a much weaker condition, $\mathcal{L}_n(\hat{\boldsymbol{\theta}}) \lesssim \Delta$ rather than $\hat{\boldsymbol{\theta}} \in \arg\min \mathcal{L}_n(\boldsymbol{\theta})$.*

Finally, in the context of the previous two results, we prove a lower bound showing statistical optimality under the imposed assumptions.

**Theorem 3.** *Suppose $d \geq 3$ and $k \leq 2^d/4$. Then, there exists a parameter class $\Theta$, and an associated function class $\mathcal{F} = \{f_{\boldsymbol{\theta}} : \boldsymbol{\theta} \in \Theta\}$ such that for any learner $\hat{\boldsymbol{\theta}}$, there exists a ground truth function $f_{\boldsymbol{\theta}^*} : \{\pm 1\}^d \to \mathbb{R}$ having a $k$-sparse polynomial representation, such that given a sufficiently large dataset of $n$ samples,*

1. *Assumptions 1(a), 1(b), 2 and 3 are satisfied with constants $L = 1$, $\mu = 0$ and $C_{n,\delta^*} \in \left[\frac{1}{2}, \frac{3}{2}\right]$ with probability at least $1 - \delta$.*

2. *$\mathbb{E}\left[\|\hat{\boldsymbol{\theta}} - \boldsymbol{\theta}^*\|_2\right] \gtrsim \sigma\sqrt{\frac{kd}{n}}$, where $\sigma^2$ denotes the noise variance in the observed labels.*

**Extensions to the case of non-unique minima.** While we focus on general parametric function classes in Theorems 1 and 2, for the specific case of DNNs, the issue of "permutation invariance" can appear. In particular, a permutation (i.e., relabeling) of the neurons in the same layer of a network can result in a different network with the same functional relationship between the input and the output. In this case, Assumptions 1(a) and 1(b) cannot hold globally for all $\boldsymbol{\theta} \in \mathbb{R}^m$, as the loss $\mathrm{Err}_n(\boldsymbol{\theta}, \boldsymbol{\theta}^*)$ and its gradient become 0 at any parameter $\boldsymbol{\theta}_\sigma^*$, where $\boldsymbol{\theta}_\sigma^*$ denotes the weight matrices corresponding to a functionally invariant permutation $\sigma$ of the neurons of the base network with parameter $\boldsymbol{\theta}^*$. However, even in this case, we can extract a guarantee from Theorem 1 and Theorem 2 by modifying the underlying assumptions slightly.

In particular, let $\boldsymbol{\Theta}^* \subset \mathbb{R}^m$ denote the set of parameters that are functionally equivalent to $\boldsymbol{\theta}^*$, in that $f_{\boldsymbol{\theta}^*} = f_{\boldsymbol{\theta}}$ for all $\theta \in \boldsymbol{\Theta}^*$. These parameters are local minimizers of $\mathrm{Err}_n(\boldsymbol{\theta}, \boldsymbol{\theta}^*)$. In particular, we define a modified RSI and QG condition, as below,

$$\langle \boldsymbol{\theta} - \boldsymbol{\theta}_{\mathrm{proj}}^*, \nabla_{\boldsymbol{\theta}}\mathrm{Err}_n(\boldsymbol{\theta}, \boldsymbol{\theta}^*)\rangle \geq C_{n,\delta}^*\|\boldsymbol{\theta} - \boldsymbol{\theta}_{\mathrm{proj}}^*\|_2^2, \qquad \text{(Modified Assumption 1(a))}$$

$$\mathrm{Err}_n(\boldsymbol{\theta}, \boldsymbol{\theta}^*) \geq C_{n,\delta}^*\|\boldsymbol{\theta} - \boldsymbol{\theta}_{\mathrm{proj}}^*\|_2^2,, \qquad \text{(Modified Assumption 1(b))}$$

where $\boldsymbol{\theta}_{\mathrm{proj}}^* = \arg\min_{\boldsymbol{\theta} \in \boldsymbol{\Theta}^*}\|\boldsymbol{\theta}^* - \boldsymbol{\theta}\|$ is the projection of $\boldsymbol{\theta}$ onto $\boldsymbol{\Theta}^*$ in some metric $\|\cdot\|$. In particular, for each $\boldsymbol{\theta} \in \boldsymbol{\Theta}^*$, let $\mathcal{K}_{\boldsymbol{\theta}}$ denote the subset of $\mathbb{R}^m$ such that for all $\boldsymbol{\theta}' \in \mathcal{K}_{\boldsymbol{\theta}}$, $\boldsymbol{\theta} = \arg\min_{\boldsymbol{\theta} \in \boldsymbol{\Theta}^*}\|\boldsymbol{\theta}^* - \boldsymbol{\theta}\|$. In other words, $\mathcal{K}_{\boldsymbol{\theta}}$ are the set of parameters which are closest to $\boldsymbol{\theta}$ among all the functionally equivalent parameters $\boldsymbol{\theta}^*$. Then, modified Assumptions 1(a) and 1(b) impose the RSI and QG conditions in a local neighborhood $(\mathcal{K}_{\boldsymbol{\theta}})$ of each $\boldsymbol{\theta} \in \boldsymbol{\Theta}^*$.

Corresponding to the new modified assumptions, a learner which returns any $\hat{\boldsymbol{\theta}}$ which is a stationary point under modified Assumption 1(a) (resp. approximate first order stationary interpolator, under modified Assumption 1(b)), guarantees to approximate $\boldsymbol{\theta}_{\mathrm{proj}}^*$, the nearest local minimizer to $\boldsymbol{\theta}$ in $\boldsymbol{\Theta}^*$. The proofs of these

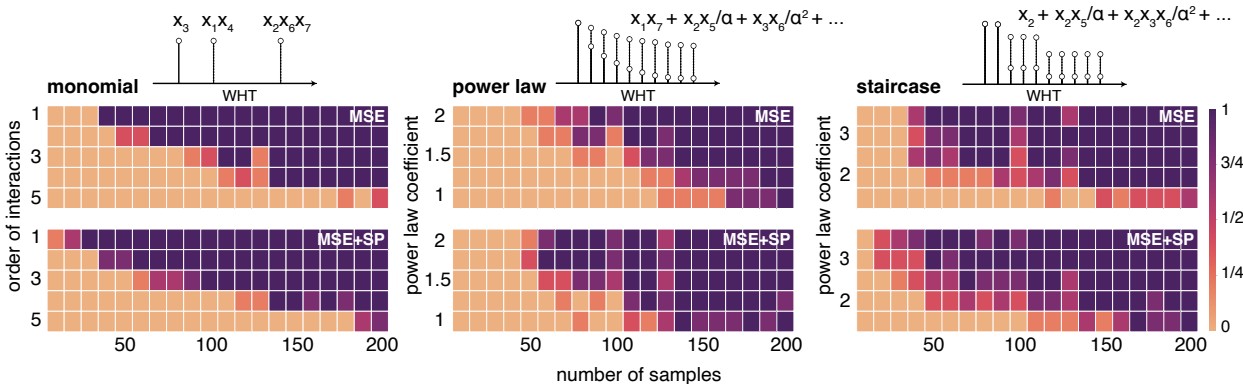

Figure 1: The plots demonstrate the generalization power of a depth-4 feed-forward neural network in learning 3 classes of sparse pseudo-Boolean functions $f(x) : \{\pm1\}^{13} \to \mathbb{R}$ using the mean squared error (MSE) loss before (first row) and after (second row) adding the spectral (SP) regularizer, as a function of the number of training samples. MONOMIAL are 1-sparse functions in Walsh-Hadamard transform (WHT) with an increasing order of interactions (1 to 5). POWER LAW are 10-sparse functions in WHT with second order interactions and coefficients set using a power law function. STAIRCASE are 18-sparse functions in WHT with 3 first order, 6 second order, and 9 third order interactions, each with an equal coefficients and set using a power law function. The heat maps show the fraction of times (among 5 random independent trials) the models generalize on unseen data with the coefficient of determination larger than $R^2 \geq 0.45$ on the test data points.

results follow identically to Theorem 1 and Theorem 2. In particular, replacing $\boldsymbol{\theta}^*$ by the parameter $\boldsymbol{\theta}^*_{\text{proj}}$, and following the same argument completes the proof of the result showing that $\boldsymbol{\theta}^*_{\text{proj}}$ (which is functionally equivalent to $\boldsymbol{\theta}^*$) can be recovered approximately.

In summary, rather than assuming that the loss $\text{Err}_n(\boldsymbol{\theta}, \boldsymbol{\theta}^*)$ is globally bowl shaped as in Assumption 1(b), an assumption which cannot hold globally for example in neural networks because of permutation invariance, it suffices to assume that the loss function is locally bowl shaped around functionally invariant permutations $\boldsymbol{\theta} \in \boldsymbol{\Theta}^*$ of the ground truth parameter $\boldsymbol{\theta}^*$. This line of reasoning can be extended to incorporate other symmetries in the function class which prevent exact parameter recovery, as in the case of permutation invariance of neural networks. The corresponding parameter recovery guarantee is also modified to be up to this symmetry.

### 4.3 Computational complexity of minimizing the regularized loss (OBJ)

In this section we discuss the computational complexity of finding stationary points of the regularized loss $\mathcal{L}_n(\boldsymbol{\theta}) + R(\boldsymbol{\theta})$ in (OBJ). Note that Aghazadeh et al. (2021) propose an algorithm for minimizing the regularized loss in (OBJ) based on an alternating minimization based approach and sparse WHT algorithms. In this section, we provide different insights on how first-order methods can be used to minimize the regularized loss, based on computing sparse WHTs.

To begin with, the cost of computing the gradient of $\mathcal{L}_n(\boldsymbol{\theta})$ for many function classes scales as $O(n \cdot \text{poly}(m))$, where $m = \dim(\boldsymbol{\theta})$. Note that we can also compute a stochastic gradient in time which scales as $O(\text{poly}(m))$ by simply computing the gradient of $(f_{\boldsymbol{\theta}}(x^i) - y^i)^2$ for a single $(x^i, y^i)$ pair. The variance of this stochastic gradient estimate largely depends on the nature of $f_{\boldsymbol{\theta}}$, as well as the training data.

On the other hand, consider the regularization term $R(\boldsymbol{\theta}) = \frac{\lambda}{\sqrt{2^d}} \|\mathbf{H} f_{\boldsymbol{\theta}}(\mathbf{X})\|_1$. In general, the complexity of exactly computing the gradient of $R$ is exponential in $d$ without any further assumptions, as it involves the summation of $2^d$ terms. While this might prove to be a challenge, empirically we observe that a two-step training process can help alleviate this issue. We first carry out *weight initialization*, running a few iterations of gradient descent on the unregularized loss, $\mathcal{L}_n(\boldsymbol{\theta})$ which generalizes moderately well, and is therefore somewhat sparse in the spectral domain. Warm-starting from this initialization and running

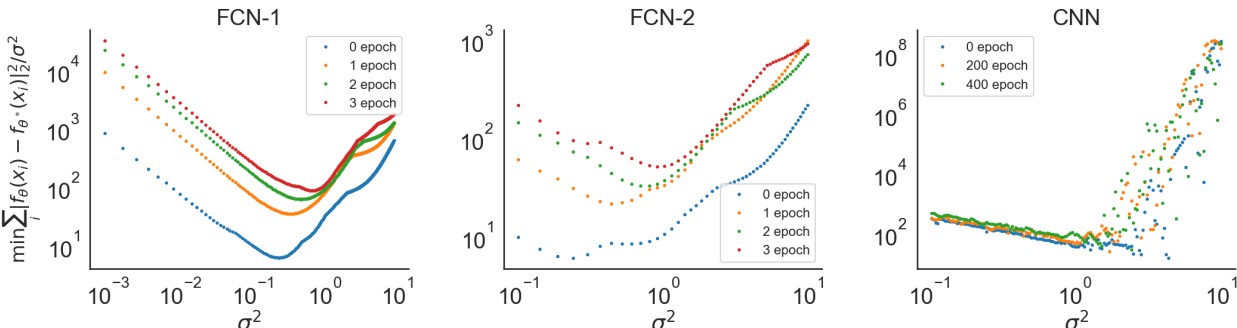

Figure 2: We estimate the empirical lower bound $\hat{C}_{n,\delta}^*$ in the quadratic growth (QG) condition by finding the minimum change in the network's output as a result of perturbations to the weights with noise drawn from the normal distribution $W \sim \mathcal{N}(0, \sigma^2)$. Experiments on 2 fully connected networks (FCNs) and a convolutional neural network (CNN) reveals empirical lower bounds.

gradient descent on the regularized loss in (OBJ), it is empirically observed that the iterates of gradient descent remain sparse throughout the training trajectory. The key implication of this result is that one can now use sparse WHT based techniques, such as Li et al. (2015) to now compute an approximation of the gradient of the regularizer at time $t$, $\nabla R(\boldsymbol{\theta}_t)$ at a computational cost scaling polynomially in $m$, and linearly in the sparsity of $f_{\boldsymbol{\theta}_t}$ in the spectral domain, which is small at each time $t$ in the second phase.

## 5 Empirical Studies

We design our experiments in a way to address these questions:

- Does SP improve generalization accuracy in learning sparse polynomials?

- When does QG hold for common DNN architectures? what is the empirical lower bound $\hat{C}_{n,\delta}^*$?

- Does SP improve generalization accuracy in real-world problems?

- How does $L_1$-regularization compare to SP-regularization in practice in terms of generalization?

**Sparse polynomials**. We compare the generalization performance of a depth-4 fully connected network with and without the SP regularization on 3 classes of sparse polynomials in Fig. 1: 1) MONOMIAL are randomly-drawn 1-sparse functions in WHT with increasing order of interactions from 1 to 5. 2) POWER LAW are randomly-drawn 10-sparse functions in WHT all with order-2 interactions and coefficients set based on a power law function with decreasing exponent. 3) STAIRCASE are randomly-drawn 18-sparse functions in the WHT with 3 first order, 6 second order, and 9 third order interactions, each with equal coefficients and set based on a power law function. These synthetic sparse polynomials are inspired by physical models for real-world pseudo-Boolean functions (e.g., protein functions (Brookes et al., 2022; Qian et al., 2001; Qin & Colwell, 2018)). We observe clear transitions in generalization power: it is harder in terms of sample cost to learn nonlinear models with higher order interactions and lower sparsity exponent (i.e., denser functions in WHT). This analysis adds a new axis to the accuracy-vs-sample-cost phase transition curves in compressed sensing (Donoho et al., 2011). SP regularization consistently improves the transition curves for generalization power as a function of order and sample cost.

**QG condition**. To empirically estimate the lower bound $\hat{C}_{n,\delta}^*$, we follow this procedure. We collect a set of training data points $(x^i, y^i)_{i=1}^{n=1000}$, initialize a DNN at $\boldsymbol{\theta}^*$ (see below for more detail), and repeatedly perturb the weights with Gaussian noise to generate $\boldsymbol{\theta}^{\text{pert},k} = \boldsymbol{\theta}^* + W_k$, where $W \sim \mathcal{N}(0, \sigma^2 I)$ is independent and normally distributed, for $k = 1, \cdots, K$. For each $k$, we compute the ratio $\sum_{i=1}^n (f_{\boldsymbol{\theta}^{\text{pert},k}}(x^i) - f_{\boldsymbol{\theta}^*}(x^i))^2 / \sigma^2$ and report the minimum value across $k \in [K]$ as a function of $\sigma^2$ in Fig. 2. Next, we train the DNN with SP regularization for certain number of epochs to arrive at a new $\boldsymbol{\theta}^*$ and repeat the same procedure

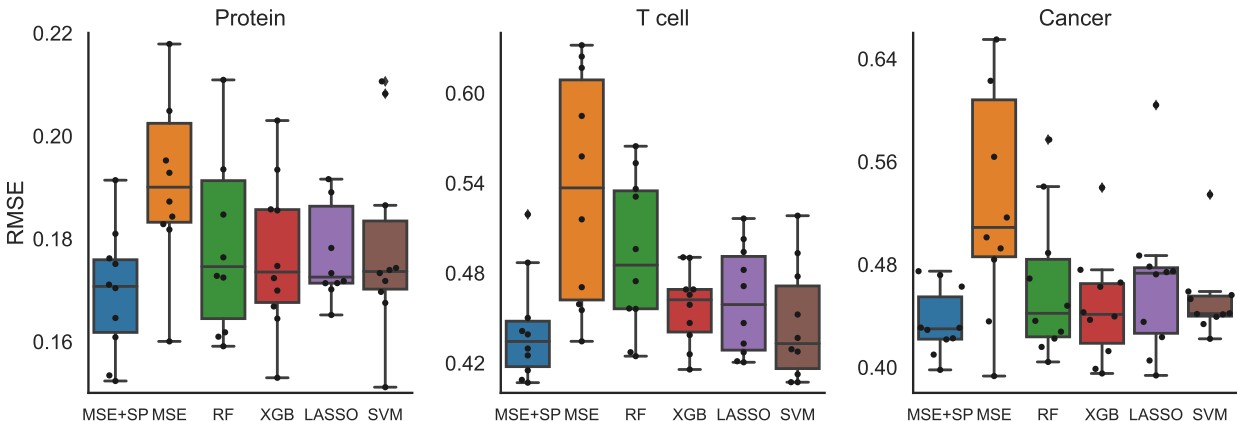

Figure 3: The plot demonstrates the generalization error of depth-4 neural networks with the mean squared error (MSE) loss before and after adding the spectral (SP) regularizer as compared to Random Forest, XGBoost, LASSO, and SVM. SP drastically improves the generalization error of neural network (reduced mean and variance) and allows for a superior performance over LASSO with $L_1$ norm regularization over the polynomial representation directly.

again to observe how the lower bound changes with training. The outputs $y$ are generated from binary vectors $x \in \{\pm 1\}^{13}$ using the (randomly-drawn) sparse polynomial $f(x) = 3x_1 + 4x_2x_3 + 5x_4x_5 + x_{12}$. We choose sufficiently small architectures compared to the number of perturbations to ensure a reliable empirical estimate for $\hat{C}^*_{n,\delta}$. Fig. 2 shows the results for Xavier-initialized, depth-2 FCNs with 222 parameters, perturbed $K = 100$ (FCN-1) and $K = 500$ (FCN-2) times, with different ranges of $\sigma^2$ (learning rate $= 5 \times 10^{-3}$). Fig. 2 also shows the result for a depth-4 CNN with 1551 parameters, initialized by a trained network using MSE loss and perturbed $K = 6000$ times (learning rate $= 10^{-3}$). The plots demonstrate that the QG condition is satisfied for these instances with empirical lower bounds 6.02 and 19.5, respectively. The bound also improves drastically with training; moreover, initializing the networks with trained models consistently improves the lower bound. See Appendix A for more detailed empirical studies and discussions of these results.

**Real world experiments**. We compare the generalization error with and without the SP regularization to standard baseline algorithms for data-scarce learning in the real-world (Fig. 3). PROTEIN is a dataset which measures the fluorescence level of $2^{13}$ protein sequences that link two variants of the *Entacmaea quadricolor* proteins different at exactly 13 amino acids (Poelwijk et al., 2019). T CELL is a dataset which measures the DNA repair outcome of T cells (average length of deletions) on 1521 sites on human genome after applying double-strand breaks (DSBs) using CRISPR (Leenay et al., 2019). CANCER is a similar dataset on 287 sites on cancer genome (Leenay et al., 2019). In the two latter datasets, a one-hot-encoded context sequence of size 20 around the DSB is used as the input to predict the DNA repair outcome. Following the low-$n$ experimental setting in (Biswas et al., 2021), we use a subset of 30 sequences drawn uniformly at random for training and validation and use the rest for testing. We repeat each experiments $10\times$ with independent random splits of the data and report the RMSE in predicting the phenotype. We initialize DNNs using Xavier (equal seeds). We use the default hyperparamters in scikit-learn for the baselines. Despite minimal hyperparameter tuning and no architecture search, SP allows for a competitive performance. In particular, the SP-regularized model outperforms LASSO which applies the $L_1$ norm penalty on the coefficients in the polynomial representation (Fig. 8 in Appendix A compares SP with different regularizers).

## 6 Summary, Discussion, and Future Vision

In the paper, we focused on learning sparse pseudo-Boolean functions with explicit regularization in the spectral domain and established the conditions on general nonlinear functions under which the stationary points which approximately interpolate the training data achieve statistically optimal generalization performance.

We demonstrated how the assumptions can be extended to nonlinear functions with multiple local minimas. Real-world experiments with neural networks demonstrated the utility of these assumptions. Naturally, some of the bounds might result in larger constants for very deep neural network (due to higher Lipschitz constants). Future works involve analyzing the algorithms that can achieve such statistical performance. It would be tempting to ask under what conditions stochastic gradient descend achieves stationary points with restricted secant and quadratic growth conditions (for which we have clear empirical evidences). Further, the statistical analysis of the computationally-efficient optimization algorithms for spectral regularization is still poorly understood. On the algorithmic side, spectral algorithms for pseudo-Boolean functions can be extended to generalized Fourier transform to accommodate a larger class of data-frugal combinatorial problems. Overall, our work provides a concrete framework to connect combinatorial algorithms with strong theoretical guarantees and nonlinear machine learning models with strong generalization power.

**Broader Impact Statement**

We do not anticipate this work to have any potential negative impact on a user of this research.

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

## A    Additional Empirical Validations

### A.1    Visualization of real-world functions in WHT

In this subsection, we visualize the combinatorial function evaluations and the WHT of the evaluation vector, that is, the left and right hand sides of the equation below:

$$\left[f(x) : x \in \{\pm 1\}^d\right]^T = \mathbf{H}\left[\alpha_z : z \in \{\pm 1\}^d\right]^T, \tag{14}$$

for the experimental data obtained from the fluorescence protein (Poelwijk et al., 2019). In addition to these plots, we have attached videos as supplemental materials where we visualize the learning trajectory of a DNN initialized using Xavier initialization under data-scarce (red) and data-sufficient (blue) regimes. DNNs start from a local minima that does not have a well-structured WHT representation, and then gets sparser with training. However, if a sufficient amount of data is not available for training, the network does not converge to a good solution. Spectral regularisation enable DNN converge to the sparse solution in the data-scarce regime.

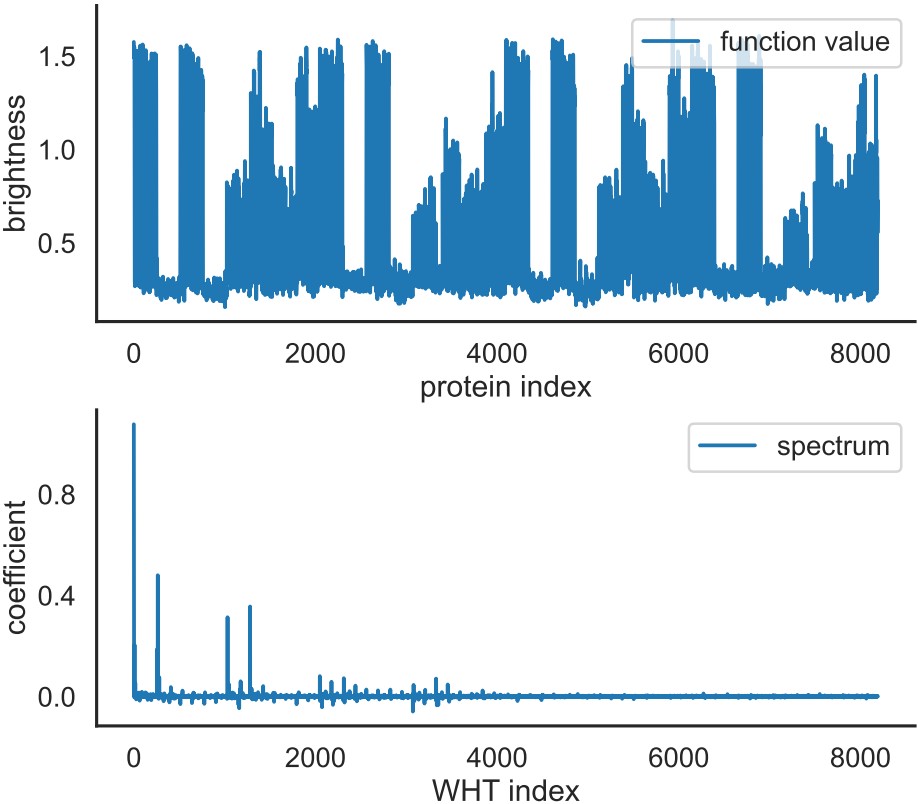

Figure 4: Combinatorial visualization of the brightness of $2^{13}$ proteins (top) and their Walsh-Hadamard transform (bottom) linking two variants of a fluorescence protein that are different in 13 locations on their amino acid chain (Poelwijk et al., 2019). The brightness is a pseudo-Boolean function which maps from $f : \{\pm 1\}^{13} \to \mathbb{R}$. The sparse spectrum reveals low and high order interactions among amino acid sites on the protein.

### A.2    Convergence: training and validation

In this subsection, we include additional empirical results which focus on the convergence properties of spectral regularization for training DNNs. The first objective we study is in validating whether stochastic gradient descent (SGD) indeed converges to approximate first order stationary interpolators (Definition 3).

The goal of these experiments are to see whether, upon running SGD, the MSE loss gets sufficiently small even when the training loss, against which stochastic gradients are computed, is augmented with spectral regularization. In Fig. 5, we plot the empirical training and validation loss of DNNs trained both with the MSE loss and the additional spectral regularization. We use a depth-4 fully connected network (learning rate= $1 \times 10^{-1}$) to train on the fluorescence protein (Poelwijk et al., 2019) dataset (also used in the main paper).

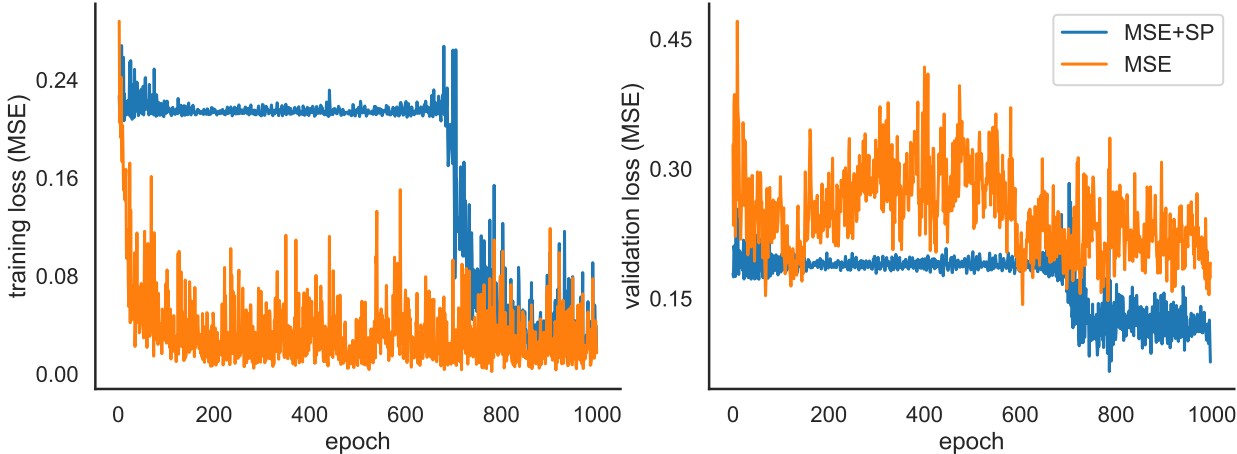

Figure 5: We plot the training and validation error in terms of the empirical mean squared error (MSE) in predicting the brightness of a fluorescence protein (Poelwijk et al., 2019) using DNNs trained with and without the spectral regularization. We use a depth-4 fully connected neural network with Xavier initialization in both cases. The plots demonstrate that 1) the training MSE eventually gets sufficiently small, even when the original loss function on which SGD is run is augmented with spectral regularization, even when network uses a random initialization and 2) spectral regularization consistently allows for a significantly better generalization gap all along the trajectory of training process.

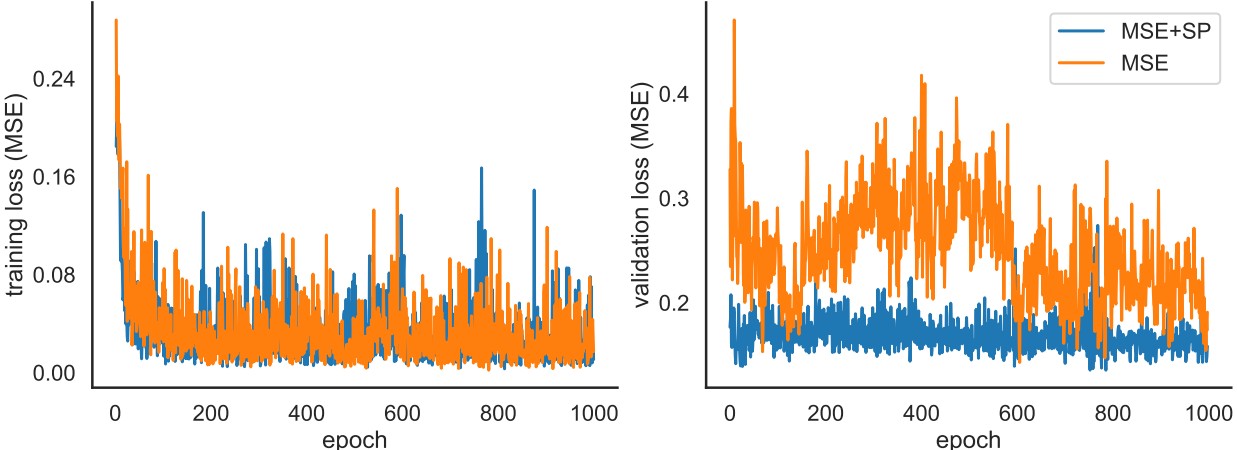

Figure 6: We repeat the experiment above with the only difference that instead of a Xavier initialization, we carry out what is known as "weight initialization" (Definition 4). Here, we train the DNN with spectral regularization (blue curve) in comparison with a DNN without spectral regularization (orange curve), but with weight initialization. The plots demonstrate that even with weight intialization, spectral regularization does not result in an increase in training, and results in a decrease in validation MSE. Weight initialization has the advantage that the QG constant encountered along the training trajectory of SGD are higher (Figure 7) compared to with a Xavier initialization

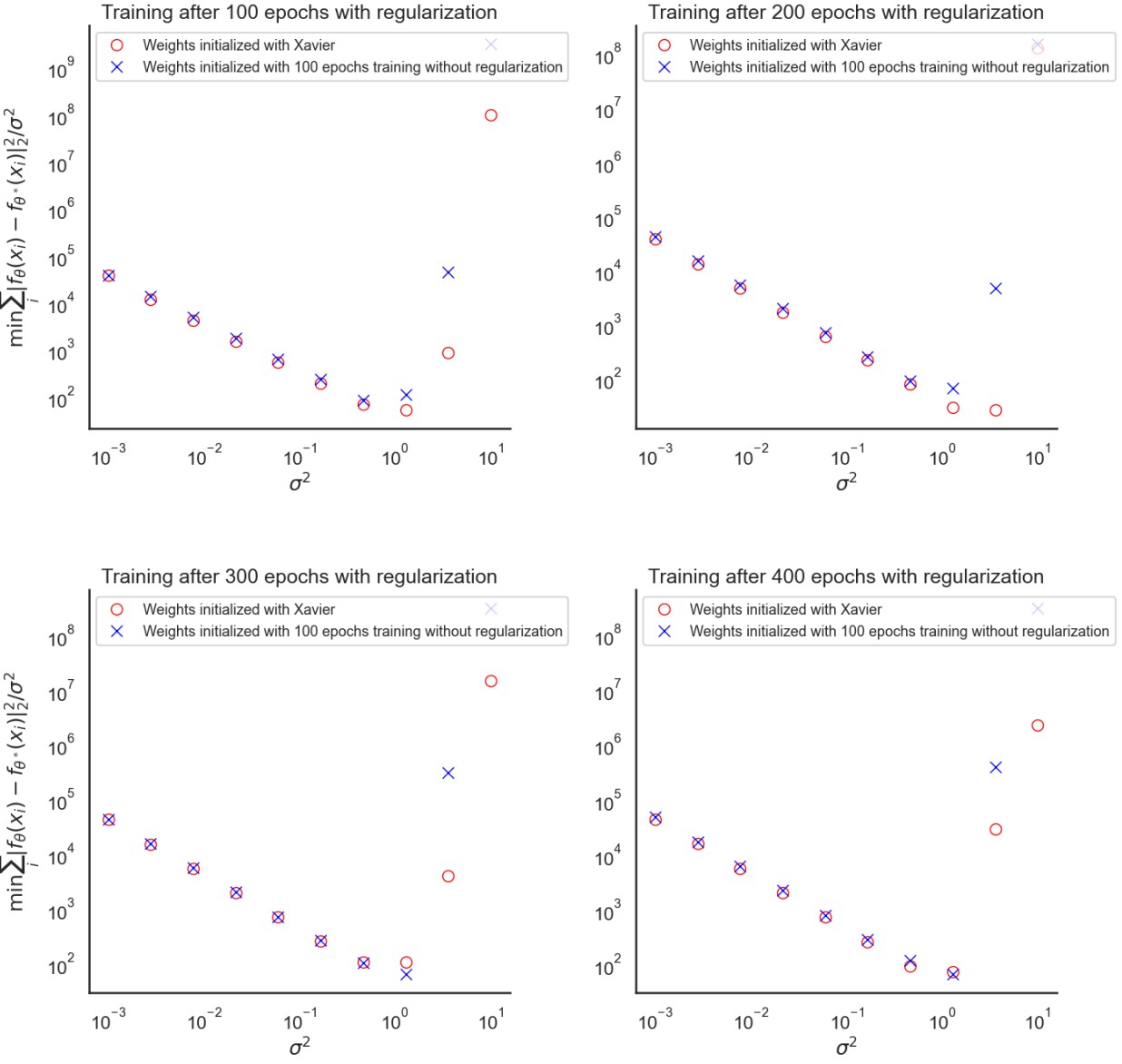

Figure 7: We plot the minimum ratio $\min_{k \in [K]} \sum_{i=1}^{n} (f_{\boldsymbol{\theta}^{\mathrm{pert},k}}(x^i) - f_{\boldsymbol{\theta}^*}(x^i))^2 / \sigma^2$ against $\sigma^2$ for 4-layer CNN models trained initialized from (i) Xavier initialization, and (ii) weight initialization, with a learning rate of $10^{-3}$. Similar to the experiments plotted in Fig. 2, we perturb $\boldsymbol{\theta}^*$, $K = 6000$ times to generate $\boldsymbol{\theta}^{\mathrm{pert},k}$.

## A.3 Validating assumption 1(b): additional plots

In this subsection, we include additional plots on empirical experiments on validating **Assumption 1(b)**. The goal of this experiment is to also provide more details about the new weight initialization method discussed in the Empirical Studies section of the main paper. Note that Assumption 1(b) requires showing that for any $\theta$,

$$\mathbb{E}_{x \sim \mathrm{Unif}(D_n)} \left[ (f_{\boldsymbol{\theta}}(x) - f_{\boldsymbol{\theta}^*}(x))^2 \right] \geq C_{n,\delta}^* \|\boldsymbol{\theta} - \boldsymbol{\theta}^*\|_2^2. \tag{15}$$

However, since it is prohibitively expensive to check this for all choices of $\theta$, and moreover since the models we consider exhibit some smoothness, we resort to checking this condition only for randomly sampled $\boldsymbol{\theta}$ around

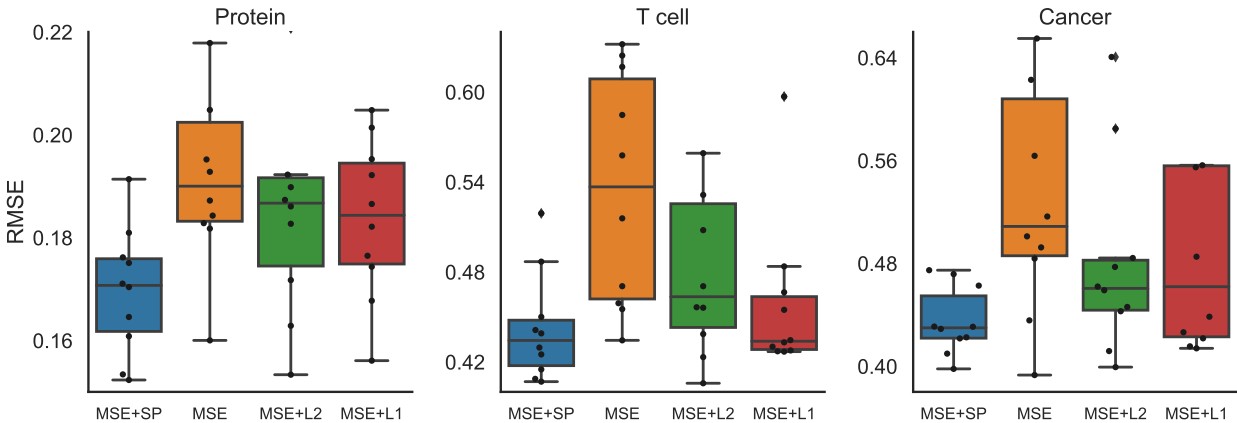

Figure 8: Comparison of the generalization error of different regularization schemes: spectral (SP), L1, and L2 norm regularization.

the reference $\boldsymbol{\theta}^*$. In particular, we collect a set of training data points $(x^i, y^i)_{i=1}^{n=1000}$ from an arbitrary sparse polynomial, $f(x) : \{\pm1\}^{13} \to \mathbb{R}$, defined as $3x_1 + 4x_2x_3 + 5x_4x_5 + x_{12}$.

First we train a DNN using SGD on the unregularized MSE, and define this as $\boldsymbol{\theta}^*$. Then, we repeatedly perturb the weights $K$ times independently with Gaussian noise to generate $\boldsymbol{\theta}^{\text{pert},k} = \boldsymbol{\theta}^* + W_k$, where $W \sim \mathcal{N}(0, \sigma^2 I)$ is independent and normally distributed, for $k = 1, \cdots, K$. By concentration of measure, under the Gaussian perturbations, note that $\|\boldsymbol{\theta}^{\text{pert},k} - \boldsymbol{\theta}^*\|_2^2$ concentrates around $M\sigma^2$, where $M$ is the number of parameters in the network. Therefore, instead of computing the ratio,

$$C_{n,\delta}^* \min_{\boldsymbol{\theta}} \frac{\mathbb{E}_{x \sim \text{Unif}(D_n)} \left[ (f_{\boldsymbol{\theta}}(x) - f_{\boldsymbol{\theta}^*}(x))^2 \right]}{\|\boldsymbol{\theta} - \boldsymbol{\theta}^*\|_2^2} \tag{16}$$

we instead approximate it by the ratio (up to scaling by $M$),

$$\min_{k \in [K]} \frac{\sum_{i=1}^n \left[ (f_{\boldsymbol{\theta}^{\text{pert},k}}(x^i) - f_{\boldsymbol{\theta}^*}(x^i))^2 \right]}{\sigma^2} \tag{17}$$

as an approximate proxy for perturbations at the scale of $\sigma$ around $\boldsymbol{\theta}^*$ in $L_\infty$ distance, which is more accurate, as $K$ grows larger. In Fig. 2, we plot the ratio in eq. (17) as a function of $\sigma$.

We repeat this experiment when $\boldsymbol{\theta}^*$ is trained starting from weight initialization, and plot the estimated ratio in Fig. 7.

**Definition 4** (Weight initialization). *Weight initialization, refers to initialization of the network by first running 100 epochs of SGD against the unregularized MSE. From this starting point, subsequently we "turn on" the regularization and run SGD against the MSE with spectral regularization.*

Each subplot shows the estimated QG ratio (eq. (17)) for models initialized with weight initialization, at the 100, 200, 300 and 400 epochs mark respectively. The plots demonstrate that our weight initialization method significantly improves the QG constant $C_{n,\delta}^*$ along sample trajectory encountered by the SGD, compared to models trained with spectral regularization starting from a random Xavier initialization.

**Remark 6.** *The guarantees hold for a specific choice of regularization parameter $\lambda$, which may also be adaptively changed over the course of optimization in the computation of gradients. In practice, the regularization parameter $\lambda$ is chosen by cross-validation.*

## A.4   Comparison to other regularization schemes

In this subsection, we compare the performance of neural network under SP regularization with other regularization schemes which directly penalize the $\ell_1$ and $\ell_2$ norm of the weights of the neural network. The goal

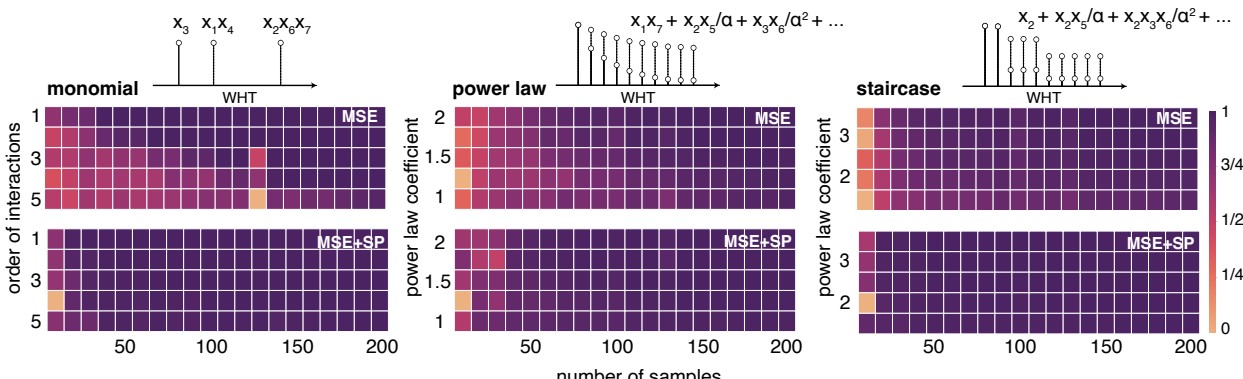

Figure 9: The plots demonstrate the generalization power of a depth-4 feed-forward neural network in learning 3 classes of sparse pseudo-Boolean functions $f(x) : \{\pm 1\}^{13} \to \mathbb{R}$ using the mean squared error (MSE) loss before (first row) and after (second row) adding the spectral (SP) regularizer, as a function of the number of training samples. MONOMIAL are 1-sparse functions in Walsh-Hadamard transform (WHT) with an increasing order of interactions (1 to 5). POWER LAW are 10-sparse functions in WHT with second order interactions and coefficients set using a power law function. STAIRCASE are 18-sparse functions in WHT with 3 first order, 6 second order, and 9 third order interactions, each with an equal coefficients and set using a power law function. The heat maps show the average of the coefficient of determination among 5 random independent trials on unseen data.

of these experiments is to show that promoting sparsity among the weights does not have the same effect as promoting sparsity in the spectral representation of neural networks. We test the generalization power of these networks using the datasets in Fig. 3 under the same experimental conditions. In Fig. 8 we demonstrate that while direct weight-regularization schemes such as $\ell_1$ and $\ell_2$ norm improve the generalization gap of neural networks, the generalization gap is significantly larger for SP regularization.

## B    Technical lemmas

In this section we outline and prove the lemmas used for the proof of the main theorems.

We first state the subgradient of the regularizer $R(\boldsymbol{\theta}) = \frac{\lambda}{\sqrt{2^d}}\|\mathbf{H}f_{\boldsymbol{\theta}}(\mathbf{X})\|_1$.

**Proposition 1.** *The subgradient of the regularizer* $R(\boldsymbol{\theta}) = \frac{\lambda}{\sqrt{2^d}}\|\mathbf{H}f_{\boldsymbol{\theta}}(\mathbf{X})\|_1$ *is,*

$$(\nabla R)(\boldsymbol{\theta}) = \left\{ \frac{\lambda}{\sqrt{2^d}} \left(\nabla f_{\boldsymbol{\theta}}(\mathbf{X})\right)\mathbf{H}z : z \in sgn(\mathbf{H}f_{\boldsymbol{\theta}}(\mathbf{X})) \right\} \tag{18}$$

*Here, for* $z \in \mathbb{R}$, $sgn(z) = \begin{cases} \{+1\} & \text{if } z > 0 \\ \{-1\} & \text{if } z < 0 \\ [-1, +1] & \text{otherwise} \end{cases}$ *and applied on a vector* $z = (z_1, \cdots, z_n)$ *in the Cartesian product of the sets* $sgn(z_1) \times \cdots \times sgn(z_n)$, *and* $\nabla f_{\boldsymbol{\theta}}(\mathbf{X})$ *is matrix with columns* $\nabla f_{\boldsymbol{\theta}}(x)$ *for each* $x \in \mathbf{X}$.

*Proof.* The proof follows by direct computation. Observe that, $\nabla\|\mathbf{H}f_{\boldsymbol{\theta}}(\mathbf{X})\|_1 = \sum_{i=1}^{2^d} \nabla|\langle e_i, \mathbf{H}f_{\boldsymbol{\theta}}(\mathbf{X})\rangle| = \sum_{i=1}^{2^d} z_i \cdot (\nabla f_{\boldsymbol{\theta}}(\mathbf{X}))\mathbf{H}^T e_i = (\nabla f_{\boldsymbol{\theta}}(\mathbf{X}))\mathbf{H}z$ where the last equation uses the symmetry of $\mathbf{H}$. $\qquad\square$

**Lemma 1.** $\|f_{\boldsymbol{\theta}^*}(\mathbf{X}) - f_{\hat{\boldsymbol{\theta}}}(\mathbf{X})\|_2 \leq 2\mu\sqrt{2^d}\|\hat{\boldsymbol{\theta}} - \boldsymbol{\theta}^*\|_2^2 + 2L\sqrt{2^d}\|\hat{\boldsymbol{\theta}} - \boldsymbol{\theta}^*\|_2$.

*Proof.* By the Lagrange form of the Taylor series expansion, for any $x$, there exists a $\boldsymbol{\theta}_x$ such that,

$$f_{\hat{\boldsymbol{\theta}}}(x) - f_{\boldsymbol{\theta}^*}(x) = (\hat{\boldsymbol{\theta}} - \boldsymbol{\theta}^*)^T \left(\nabla^2 f_{\boldsymbol{\theta}_x}(x)\right)(\hat{\boldsymbol{\theta}} - \boldsymbol{\theta}^*) + \left\langle \nabla f_{\boldsymbol{\theta}^*}(x), \hat{\boldsymbol{\theta}} - \boldsymbol{\theta}^* \right\rangle \tag{19}$$

Then, for each $x \in \mathbf{X}$,

$$\left(f_{\hat{\boldsymbol{\theta}}}(x) - f_{\boldsymbol{\theta}^*}(x)\right)^2 \leq 2\mu^2 \|\hat{\boldsymbol{\theta}} - \boldsymbol{\theta}^*\|_2^4 + 2\left\langle \nabla f_{\theta^*}(x), \hat{\boldsymbol{\theta}} - \boldsymbol{\theta}^* \right\rangle^2 \tag{20}$$

Summing over $x \in \mathbf{X}$ results in,

$$\|\nu\|_2^2 = \|f_{\hat{\boldsymbol{\theta}}}(\mathbf{X}) - f_{\boldsymbol{\theta}^*}(\mathbf{X})\|_2^2 \leq \mu^2 2^{d+1} \|\hat{\boldsymbol{\theta}} - \boldsymbol{\theta}^*\|_2^4 + L^2 2^{d+1} \|\hat{\boldsymbol{\theta}} - \boldsymbol{\theta}^*\|_2^2. \tag{21}$$

where recall the assumption, $\frac{1}{2^d} \sum_{x \in \mathbf{X}} \nabla f_{\theta^*}(x)(\nabla f_{\theta^*}(x))^T \preceq L^2$.

$\square$

**Lemma 2.** $\max_{v:\|v\|_\infty \leq 1} \|\mathbf{H}v\|_1 \leq (2d+1)\sqrt{2^d}$.

*Proof.* Note that $\max_{v:\|v\|_\infty \leq 1} \|\mathbf{H}v\|_1 = \max_{v \in \{-1,+1\}^{2^d}} \|\mathbf{H}v\|_1$, by (Rohn, 2000, Proposition 1). Therefore, it suffices to maximize over $v \in \{-1,+1\}^{2^d}$. It is known from Figula. & Kvaratskhelia (2015) that $\hat{\varrho}^{(d)} \leq d2^d$ where $\hat{\varrho}^{(d)} \triangleq \max_{m \in [2^d]} \sqrt{2^d} \|\Phi \mathbf{H} \mathbf{1}_m\|_1$ where $\Phi$ is an arbitrary set of vectors from the unit $\|\cdot\|_1$ ball in $\mathbb{R}^{2^d}$, $\mathbf{1}_m$ is the vector with the first $m$ entries 1 and the remaining entries 0. Since $\Phi$ can be chosen as any permutation matrix, this result implies that $\max_{v \in \{0,1\}^{2^d}} \|\mathbf{H}v\|_1 \leq d2^d$. To compare with the previous statement, note that the number of non-zeros in $v$ here equals $m$ in the optimization problem defining $\hat{\varrho}^{(d)}$. Then, we have that,

$$\max_{v \in \{-1,+1\}^{2^d}} \|\mathbf{H}v\|_1 = \max_{v \in \{0,1\}^{2^d}} \|\mathbf{H}(2v - \mathbf{1})\|_1 \tag{22}$$

$$\overset{(i)}{\leq} 2\|\mathbf{H}v\|_1 + \|\mathbf{H}\mathbf{1}\|_1 \tag{23}$$

$$\leq 2d\sqrt{2^d} + \sqrt{2^d}, \tag{24}$$

where $(i)$ follows by triangle inequality. $\square$

**Lemma 3.** *Under Assumptions 1(a) and 2, for any $\boldsymbol{\theta} \in \mathbb{R}^d$ and any subgradient $G \in (\nabla R)(\boldsymbol{\theta})$, under the event $\mathcal{E}_{rsi} : \langle \boldsymbol{\theta} - \boldsymbol{\theta}^*, \mathrm{Err}_n(\boldsymbol{\theta}, \boldsymbol{\theta}^*) \rangle \geq C_{n,\delta}^* \|\boldsymbol{\theta} - \boldsymbol{\theta}^*\|_2^2$ which happens with probability $\geq 1 - \delta$,*

$$\left\langle \boldsymbol{\theta} - \boldsymbol{\theta}^*, \nabla \mathcal{L}_n(\boldsymbol{\theta}) + G \right\rangle - 2\mathbb{E}_{x \sim \mathrm{Unif}(D_n)} \left[ (f_{\boldsymbol{\theta}^*}(x) - y(x))) \langle \boldsymbol{\theta} - \boldsymbol{\theta}^*, \nabla f_{\boldsymbol{\theta}}(x) \rangle \right] \tag{25}$$

$$\geq \left( C_{n,\delta}^* - (2d+1)\lambda\mu \right) \|\boldsymbol{\theta} - \boldsymbol{\theta}^*\|_2^2 + R(\boldsymbol{\theta}) - R(\boldsymbol{\theta}^*) \tag{26}$$

*Proof.* Observe that,

$$\langle \boldsymbol{\theta} - \boldsymbol{\theta}^*, \nabla \mathcal{L}_n(\boldsymbol{\theta}) \rangle = 2\mathbb{E}_{x \sim \mathrm{Unif}(D_n)} \left[ (f_{\boldsymbol{\theta}}(x) - y(x)) \langle \boldsymbol{\theta} - \boldsymbol{\theta}^*, \nabla f_{\boldsymbol{\theta}}(x) \rangle \right] \tag{27}$$

Plugging this in below,

$$\langle \boldsymbol{\theta} - \boldsymbol{\theta}^*, \nabla \mathcal{L}_n(\boldsymbol{\theta}) \rangle - 2\mathbb{E}_{x \sim \mathrm{Unif}(D_n)} \left[ (f_{\boldsymbol{\theta}^*}(x) - y(x))) \langle \boldsymbol{\theta} - \boldsymbol{\theta}^*, \nabla f_{\boldsymbol{\theta}}(x) \rangle \right] \tag{28}$$

$$= 2\mathbb{E}_{x \sim \mathrm{Unif}(D_n)} \left[ (f_{\boldsymbol{\theta}}(x) - y(x)) \langle \boldsymbol{\theta} - \boldsymbol{\theta}^*, \nabla f_{\boldsymbol{\theta}}(x) \rangle - (f_{\boldsymbol{\theta}^*}(x) - y(x)) \langle \boldsymbol{\theta} - \boldsymbol{\theta}^*, \nabla f_{\boldsymbol{\theta}}(x) \rangle \right] \tag{29}$$

$$= 2\mathbb{E}_{x \sim \mathrm{Unif}(D_n)} \left[ (f_{\boldsymbol{\theta}}(x) - f_{\boldsymbol{\theta}^*}(x)) \langle \boldsymbol{\theta} - \boldsymbol{\theta}^*, \nabla f_{\boldsymbol{\theta}}(x) \rangle \right] \tag{30}$$

$$= \left\langle \boldsymbol{\theta} - \boldsymbol{\theta}^*, \nabla \mathbb{E}_{x \sim \mathrm{Unif}(D_n)} \left[ (f_{\boldsymbol{\theta}}(x) - f_{\boldsymbol{\theta}^*}(x))^2 \right] \right\rangle \tag{31}$$

$$\geq C_{n,\delta}^* \|\boldsymbol{\theta} - \boldsymbol{\theta}^*\|_2^2 \tag{32}$$

where the last inequality follows by the RSI condition imposed on $\mathrm{Err}_n(\boldsymbol{\theta}, \boldsymbol{\theta}^*)$ in Assumption 1(a). Putting together eq. (32) with Lemma 5, uner the event $\mathcal{E}_{\mathrm{rsi}}$,

$$\langle \boldsymbol{\theta} - \boldsymbol{\theta}^*, \nabla \mathcal{L}_n(\boldsymbol{\theta}) + G \rangle - 2\mathbb{E}_{x \sim \mathrm{Unif}(D_n)} \left[ (f_{\boldsymbol{\theta}^*}(x) - y(x))) \langle \boldsymbol{\theta} - \boldsymbol{\theta}^*, \nabla f_{\boldsymbol{\theta}}(x) \rangle \right] \tag{33}$$

$$\geq C_{n,\delta}^* \|\boldsymbol{\theta} - \boldsymbol{\theta}^*\|_2^2 + R(\boldsymbol{\theta}) - R(\boldsymbol{\theta}^*) - (2d+1)\lambda\mu \|\boldsymbol{\theta} - \boldsymbol{\theta}^*\|_2^2 \tag{34}$$

This completes the proof. $\square$

In the remaining proofs, we will assume a lower bound on the regularization parameter to ensure that the regularization actually plays a role. In particular,

**Condition 1.** *Assume that the regularization parameter $\lambda$ satisfies,*

$$\lambda \geq 12\sigma\sqrt{\frac{d + \log(1/\delta)}{n}}. \tag{35}$$

**Lemma 4.** *Suppose $\lambda$ satisfies the lower bound in Condition 1. Define $\mathcal{E}_2$ as the event that, for all $\boldsymbol{\theta} \in \mathbb{R}^d$,*

$$\mathbb{E}_{x\sim\text{Unif}(D_n)}\left[(f_{\boldsymbol{\theta}^*}(x) - y(x)))\langle\boldsymbol{\theta}^* - \boldsymbol{\theta}, \nabla f_{\boldsymbol{\theta}}(x)\rangle\right] \tag{36}$$

$$\leq \frac{\lambda}{4}\sqrt{\frac{1}{2^d}}\|\mathbf{H}(f_{\boldsymbol{\theta}^*}(\mathbf{X}) - f_{\boldsymbol{\theta}}(\mathbf{X}))\|_1 + \frac{\lambda\mu(2d+1)}{4}\|\boldsymbol{\theta} - \boldsymbol{\theta}^*\|_2^2. \tag{37}$$

*Then, $\Pr(\mathcal{E}_2) \geq 1 - \delta$.*

*Proof.* For each sample $x' \in D_n$, define the noise in the sample as $z(x') = f_{\boldsymbol{\theta}^*}(x') - y(x')$. Likewise, define the noise vector $\mathbf{z}_{D_n} = \{\mathbb{E}_{x\sim\text{Unif}(D_n)}[z(x')\mathbb{I}(x' = x)] : x \in [2^d]\}$. Note that the coordinates of $\mathbf{z}_{D_n}$ corresponding to inputs unobserved in the dataset $D_n$ are 0.

By Taylor series expanding, $f_{\boldsymbol{\theta}^*}(x) - f_{\boldsymbol{\theta}}(x) = \langle\boldsymbol{\theta}^* - \boldsymbol{\theta}, \nabla f_{\boldsymbol{\theta}}(x)\rangle + (\boldsymbol{\theta}^* - \boldsymbol{\theta})^T(\nabla^2 f_{\boldsymbol{\theta}_x}(x))(\boldsymbol{\theta}^* - \boldsymbol{\theta})$ for some $\boldsymbol{\theta}_x \in \text{conv}(\{\boldsymbol{\theta}, \boldsymbol{\theta}^*\})$. Therefore,

$$\mathbb{E}_{x\sim\text{Unif}(D_n)}\left[(f_{\boldsymbol{\theta}^*}(x) - y(x)))\langle\boldsymbol{\theta}^* - \boldsymbol{\theta}, \nabla f_{\boldsymbol{\theta}}(x)\rangle\right] \tag{38}$$

$$= \mathbb{E}_{x\sim\text{Unif}(D_n)}\left[(f_{\boldsymbol{\theta}^*}(x) - y(x)))\left(f_{\boldsymbol{\theta}^*}(x) - f_{\boldsymbol{\theta}}(x) - (\boldsymbol{\theta} - \boldsymbol{\theta}^*)^T(\nabla^2 f_{\boldsymbol{\theta}_x}(x))(\boldsymbol{\theta} - \boldsymbol{\theta}^*)\right)\right] \tag{39}$$

$$= \langle\mathbf{z}_{D_n}, f_{\boldsymbol{\theta}^*}(\mathbf{X}) - f_{\boldsymbol{\theta}}(\mathbf{X}) - \mathbf{A}(\boldsymbol{\theta}, \boldsymbol{\theta}^*)\rangle, \tag{40}$$

where $\mathbf{A}(\boldsymbol{\theta}, \boldsymbol{\theta}^*)$ denotes the vector $\{(\boldsymbol{\theta}^* - \boldsymbol{\theta})^T\nabla^2 f_{\boldsymbol{\theta}_x}(x)(\boldsymbol{\theta}^* - \boldsymbol{\theta}) : x \in [2^d]\}$.

By an application of Holder's inequality and triangle inequality of the $L_1$-norm, we have,

$$\mathbb{E}_{x\sim\text{Unif}(D_n)}\left[(f_{\boldsymbol{\theta}^*}(x) - y(x)))\langle\boldsymbol{\theta}^* - \boldsymbol{\theta}, \nabla f_{\boldsymbol{\theta}}(x)\rangle\right] \tag{41}$$

$$\leq \|\mathbf{H}\mathbf{z}_{D_n}\|_\infty\|\mathbf{H}(f_{\boldsymbol{\theta}^*}(\mathbf{X}) - f_{\hat{\boldsymbol{\theta}}}(\mathbf{X}))\|_1 + \|\mathbf{H}\mathbf{z}_{D_n}\|_\infty\|\mathbf{H}\mathbf{A}(\boldsymbol{\theta}, \boldsymbol{\theta}^*)\|_1 \tag{42}$$

Note that for each fixed row $i \in [2^d]$, $\langle\mathbf{H}_i, \mathbf{z}_{D_n}\rangle = \sum_{j\in 2^d}\mathbf{H}_{ij}\mathbf{z}_{D_n}(j)$. Note that the coordinates of $\mathbf{z}_{D_n}$ are independently distributed and subgaussian. Therefore, by Bernstein's inequality,

$$\Pr\left(\langle\mathbf{H}_i, \mathbf{z}_{D_n}\rangle \geq 3\sqrt{\frac{\left(\sum_{j\in[2^d]}\text{Var}(\mathbf{z}_{D_n}(j))\right)\log(1/\delta)}{2^d}}\right) \leq \delta \tag{43}$$

Note that the coordinate of $\mathbf{z}_{D_n}$ labelled by $x \in \{\pm 1\}^d$, $\mathbb{E}_{x\sim\text{Unif}(D_n)}[z(x')\mathbb{I}(x' = x)]$, is the sum of $D_n(x)$ independent $\mathcal{N}(0, \sigma^2)$ Gaussians scaled by $1/n$, where $D_n(x)$, defined as the number of times $x$ is sampled in $D_n$. Therefore, $\text{Var}(\mathbf{z}_{D_n}(i)) = \frac{\sigma^2 D_n(x)}{n^2}$. By union bounding over the $2^d$ rows of $\mathbf{H}$, with probability $\geq 1 - \delta$,

$$\Pr\left(\|\mathbf{H}\mathbf{z}_{D_n}\|_\infty \geq 3\sqrt{\frac{\left(\sum_{x\in[2^d]}\frac{\sigma^2 D_n(x)}{n^2}\right)\log(2^d/\delta)}{2^d}}\right) \leq \delta. \tag{44}$$

Note that $\sum_{x\in[2^d]}D_n(x) = n$, and therefore, with probability $\geq 1 - \delta$, the event $\mathcal{E}_1$, defined below, is satisfied,

$$\mathcal{E}_1 : \|\mathbf{H}\mathbf{z}_{D_n}\|_\infty \leq 3\sigma\sqrt{\frac{1}{2^d}}\sqrt{\frac{d + \log(1/\delta)}{n}} \leq \frac{\lambda}{4\sqrt{2^d}}. \tag{45}$$

where the last inequality follows by the assumption on $\lambda$ in Condition 1.

Note that $\|\mathbf{A}(\boldsymbol{\theta}, \boldsymbol{\theta}^*)\|_\infty \leq \mu\|\boldsymbol{\theta} - \boldsymbol{\theta}^*\|_2^2$. Thus, the term $\|\mathbf{H}\mathbf{A}(\boldsymbol{\theta}, \boldsymbol{\theta}^*)\|_1$ can be upper bounded by $\sup_{v:\|v\|_\infty \leq \mu\|\boldsymbol{\theta}-\boldsymbol{\theta}^*\|_2^2} \|\mathbf{H}v\|_1$. This itself can be further upper bounded using Lemma 2 by $\mu\|\boldsymbol{\theta}-\boldsymbol{\theta}^*\|_2^2(2d+1)\sqrt{2^d}$.

All in all, combining this argument and eq. (45) with eq. (42), under the event $\mathcal{E}_1$, with probability $\geq 1 - \delta$,

$$\mathbb{E}_{x\sim\mathrm{Unif}(D_n)}\left[(f_{\boldsymbol{\theta}^*}(x) - y(x)))\langle\boldsymbol{\theta}^* - \boldsymbol{\theta}, \nabla f_{\boldsymbol{\theta}}(x)\rangle\right] \tag{46}$$

$$\leq \frac{\lambda}{4}\sqrt{\frac{1}{2^d}}\left\|\mathbf{H}\left(f_{\boldsymbol{\theta}^*}(\mathbf{X}) - f_{\hat{\boldsymbol{\theta}}}(\mathbf{X})\right)\right\|_1 + \frac{\lambda\mu(2d+1)}{4}\|\boldsymbol{\theta} - \boldsymbol{\theta}^*\|_2^2. \tag{47}$$

$\square$

**Lemma 5.** *Under the Lipschitz gradient condition, Assumption 2, consider any $\boldsymbol{\theta} \in \mathbb{R}^d$ and any subgradient $G \in (\nabla R)(\boldsymbol{\theta})$. Then,*

$$\langle\boldsymbol{\theta} - \boldsymbol{\theta}^*, G\rangle \geq R(\boldsymbol{\theta}) - R(\boldsymbol{\theta}^*) - (2d+1)\lambda\mu\|\boldsymbol{\theta} - \boldsymbol{\theta}^*\|_2^2. \tag{48}$$

*Proof.* Recall that the regularization function $R(\boldsymbol{\theta})$ is defined as $\frac{\lambda}{\sqrt{2^d}}\|\mathbf{H}f_{\boldsymbol{\theta}}(\mathbf{X})\|_1$. Consider any subgradient $G \in (\nabla R)(\boldsymbol{\theta})$ of the regularization function $R(\boldsymbol{\theta})$. By Proposition 1, this is of the form $\frac{\lambda}{\sqrt{2^d}}(\nabla f_{\boldsymbol{\theta}}(\mathbf{X}))\mathbf{H}z$, where $z \in \mathsf{sgn}(\mathbf{H}f_{\boldsymbol{\theta}}(\mathbf{X}))$. In particular, using this representation, we have the equation,

$$\langle\boldsymbol{\theta} - \boldsymbol{\theta}^*, G\rangle = \frac{\lambda}{\sqrt{2^d}}(\boldsymbol{\theta} - \boldsymbol{\theta}^*)^T(\nabla f_{\boldsymbol{\theta}}(\mathbf{X}))\mathbf{H}z. \tag{49}$$

Since $f_{\boldsymbol{\theta}}(x)$ is in general non-linear in $\boldsymbol{\theta}$, we can relate $(\boldsymbol{\theta} - \boldsymbol{\theta}^*)^T\nabla f_{\boldsymbol{\theta}}(x)$ to $f_{\boldsymbol{\theta}}(x) - f_{\boldsymbol{\theta}^*}(x)$ by using a Taylor series expansion. In particular, for each $x \in [2^d]$ and each $\boldsymbol{\theta}$ and $\boldsymbol{\theta}^*$, by the Lagrange form of the Taylor series expansion, there exists a $\boldsymbol{\theta}_x \in \mathsf{conv}(\{\boldsymbol{\theta}, \boldsymbol{\theta}^*\})$ such that $f_{\boldsymbol{\theta}^*}(x) - f_{\boldsymbol{\theta}}(x) = \langle\nabla f_{\boldsymbol{\theta}}(x), \boldsymbol{\theta}^* - \boldsymbol{\theta}\rangle + (\boldsymbol{\theta}^* - \boldsymbol{\theta})^T\nabla^2 f_{\boldsymbol{\theta}_x}(x)(\boldsymbol{\theta}^* - \boldsymbol{\theta})$. In addition, note from Assumption 2 that $-\mu I \preceq \nabla^2 f_{\boldsymbol{\theta}}(x) \preceq \mu I$. This results in the following set of inequalities,

$$\langle\boldsymbol{\theta} - \boldsymbol{\theta}^*, G\rangle = \frac{\lambda}{\sqrt{2^d}}\begin{bmatrix} f_{\boldsymbol{\theta}}(x) - f_{\boldsymbol{\theta}^*}(x) + (\boldsymbol{\theta} - \boldsymbol{\theta}^*)^T\nabla^2 f_{\boldsymbol{\theta}_x}(x)(\boldsymbol{\theta} - \boldsymbol{\theta}^*) \\ \vdots \end{bmatrix}_{x\in[2^d]}\mathbf{H}z \tag{50}$$

$$\geq \frac{\lambda}{\sqrt{2^d}}(f_{\boldsymbol{\theta}}(\mathbf{X}) - f_{\boldsymbol{\theta}^*}(\mathbf{X}))^T\mathbf{H}z - \frac{\lambda}{\sqrt{2^d}}\mu\|\boldsymbol{\theta} - \boldsymbol{\theta}^*\|_2^2\|\mathbf{H}z\|_1 \tag{51}$$

$$\geq \frac{\lambda}{\sqrt{2^d}}(\mathbf{H}f_{\boldsymbol{\theta}}(\mathbf{X}) - \mathbf{H}f_{\boldsymbol{\theta}^*}(\mathbf{X}))^T z - (2d+1)\lambda\mu\|\boldsymbol{\theta} - \boldsymbol{\theta}^*\|_2^2 \tag{52}$$

where the last inequality follows from Lemma 2, where we show that for any vector $v : \|v\|_\infty \leq 1$ (a condition which is satisfied by the sign vector $z$), $\|\mathbf{H}v\|_1 \leq (2d+1)\sqrt{2^d}$.

A naive attempt to prove such a bound turns out to result in a loose bound. Indeed, $\|\mathbf{H}v\|_1 \leq \sqrt{2^d}\|\mathbf{H}v\|_2 = \sqrt{2^d}\|v\|_2 \leq \sqrt{2^d}\sqrt{2^d} = 2^d$. The improvement of one of the $\sqrt{2^d}$ factors to $(2d+1)$ turns to be quite a deep mathematical fact, and we invoke a result of Figula. & Kvaratskhelia (2015) to prove this result in Lemma 2.

Finally, using the convexity of $\|\cdot\|_1$, for any $z \in \mathsf{sgn}(\mathbf{H}f_{\boldsymbol{\theta}}(\mathbf{X}))$,

$$(\mathbf{H}f_{\boldsymbol{\theta}}(\mathbf{X}) - \mathbf{H}f_{\boldsymbol{\theta}^*}(\mathbf{X}))^T z = \|\mathbf{H}f_{\boldsymbol{\theta}}(\mathbf{X})\|_1 - (\mathbf{H}f_{\boldsymbol{\theta}^*}(\mathbf{X}))^T z \tag{53}$$

$$\geq \|\mathbf{H}f_{\boldsymbol{\theta}}(\mathbf{X})\|_1 - \sup_{v:\|v\|_\infty\leq 1}\langle v, \mathbf{H}f_{\boldsymbol{\theta}^*}(\mathbf{X})\rangle \tag{54}$$

$$\geq \|\mathbf{H}f_{\boldsymbol{\theta}}(\mathbf{X})\|_1 - \|\mathbf{H}f_{\boldsymbol{\theta}^*}(\mathbf{X})\|_1 \tag{55}$$

Combining this with eq. (52) and using the definition of $R(\boldsymbol{\theta})$ completes the proof. $\square$

**Lemma 6.** *Under Assumption 2, for any $\boldsymbol{\theta} \in \mathbb{R}^m$ and any subgradient $G \in (\nabla R)(\boldsymbol{\theta})$,*

$$\left\langle \boldsymbol{\theta} - \boldsymbol{\theta}^*, \nabla \mathcal{L}_n(\boldsymbol{\theta}) + G \right\rangle + 2\mathbb{E}_{x \sim \text{Unif}(D_n)} \left[ (f_{\boldsymbol{\theta}^*}(x) - y(x))) \left( f_{\boldsymbol{\theta}^*}(x) - f_{\boldsymbol{\theta}}(x) \right) \right] \tag{56}$$

$$\geq 2\text{Err}_n(\boldsymbol{\theta}, \boldsymbol{\theta}^*) + R(\boldsymbol{\theta}) - R(\boldsymbol{\theta}^*) - \left( (2d+1)\lambda\mu + 2\mu\sqrt{\mathcal{L}_n(\boldsymbol{\theta})} \right) \|\boldsymbol{\theta} - \boldsymbol{\theta}^*\|_2^2 \tag{57}$$

*Proof.* The proof of this result builds on the analysis in Lemma 3. Observe that,

$$\langle \boldsymbol{\theta} - \boldsymbol{\theta}^*, \nabla \mathcal{L}_n(\boldsymbol{\theta}) \rangle = \mathbb{E}_{x \sim \text{Unif}(D_n)} \left[ 2(f_{\boldsymbol{\theta}}(x) - y(x)) \langle \boldsymbol{\theta} - \boldsymbol{\theta}^*, \nabla f_{\boldsymbol{\theta}}(x) \rangle \right] \tag{58}$$

For each $x \in [2^d]$ and each $\boldsymbol{\theta}$ and $\boldsymbol{\theta}^*$, there exists a $\boldsymbol{\theta}_x \in \text{conv}(\{\boldsymbol{\theta}, \boldsymbol{\theta}^*\})$ such that $f_{\boldsymbol{\theta}}(x) - f_{\boldsymbol{\theta}^*}(x) = \langle \nabla f_{\boldsymbol{\theta}}(x), \boldsymbol{\theta} - \boldsymbol{\theta}^* \rangle - (\boldsymbol{\theta} - \boldsymbol{\theta}^*)^T \nabla^2 f_{\boldsymbol{\theta}_x}(x)(\boldsymbol{\theta} - \boldsymbol{\theta}^*)$. Plugging this in, and using the assumption that $-\mu I \preceq \nabla^2 f_{\boldsymbol{\theta}}(x) \preceq \mu I$,

$$\langle \boldsymbol{\theta} - \boldsymbol{\theta}^*, \nabla \mathcal{L}_n(\boldsymbol{\theta}) \rangle - 2\mathbb{E}_{x \sim \text{Unif}(D_n)} \left[ (f_{\boldsymbol{\theta}^*}(x) - y(x))) \left( f_{\boldsymbol{\theta}}(x) - f_{\boldsymbol{\theta}^*}(x) \right) \right] \tag{59}$$

$$= \mathbb{E}_{x \sim \text{Unif}(D_n)} \left[ 2(f_{\boldsymbol{\theta}}(x) - y(x)) \langle \boldsymbol{\theta} - \boldsymbol{\theta}^*, \nabla f_{\boldsymbol{\theta}}(x) \rangle - 2(f_{\boldsymbol{\theta}^*}(x) - y(x))(f_{\boldsymbol{\theta}}(x) - f_{\boldsymbol{\theta}^*}(x)) \right] \tag{60}$$

$$\geq \mathbb{E}_{x \sim \text{Unif}(D_n)} \left[ 2(f_{\boldsymbol{\theta}}(x) - y(x))(f_{\boldsymbol{\theta}}(x) - f_{\boldsymbol{\theta}^*}(x)) - 2(f_{\boldsymbol{\theta}^*}(x) - y(x))(f_{\boldsymbol{\theta}}(x) - f_{\boldsymbol{\theta}^*}(x)) \right] \tag{61}$$

$$\quad - 2\mu\|\boldsymbol{\theta} - \boldsymbol{\theta}^*\|_2^2 \, \mathbb{E}_{x \sim \text{Unif}(D_n)} \left[ |f_{\boldsymbol{\theta}}(x) - y(x)| \right] \tag{62}$$

$$\geq 2\mathbb{E}_{x \sim \text{Unif}(D_n)} \left[ (f_{\boldsymbol{\theta}}(x) - f_{\boldsymbol{\theta}^*}(x))^2 \right] - 2\mu\|\boldsymbol{\theta} - \boldsymbol{\theta}^*\|_2^2 \left( \sqrt{\mathcal{L}_n(\boldsymbol{\theta})} \right) \tag{63}$$

where the last inequality follows by Jensen's inequality.

Putting together Lemma 5 and eq. (63), for any subgradient $G \in (\nabla R)(\boldsymbol{\theta})$, we have,

$$\left\langle \boldsymbol{\theta} - \boldsymbol{\theta}^*, \nabla \mathcal{L}_n(\boldsymbol{\theta}) + G \right\rangle - 2\mathbb{E}_{x \sim \text{Unif}(D_n)} \left[ (f_{\boldsymbol{\theta}^*}(x) - y(x))) \left( f_{\boldsymbol{\theta}}(x) - f_{\boldsymbol{\theta}^*}(x) \right) \right] \tag{64}$$

$$\geq 2\mathbb{E}_{x \sim \text{Unif}(D_n)} \left[ (f_{\boldsymbol{\theta}}(x) - f_{\boldsymbol{\theta}^*}(x))^2 \right] + \frac{\lambda}{\sqrt{2^d}} \left( \|\mathbf{H} f_{\boldsymbol{\theta}}(\mathbf{X})\|_1 - \|\mathbf{H} f_{\boldsymbol{\theta}^*}(\mathbf{X})\|_1 \right) \tag{65}$$

$$\quad - \left( (2d+1)\lambda\mu + 2\mu\sqrt{\mathcal{L}_n(\boldsymbol{\theta})} \right) \|\boldsymbol{\theta} - \boldsymbol{\theta}^*\|_2^2 \tag{66}$$

This completes the proof. $\qquad\square$

**Lemma 7.** *Suppose the regularization parameter satisfies the lower bound in Condition 1. Define $\mathcal{E}_3$ as the event that for all $\boldsymbol{\theta} \in \mathbb{R}^m$,*

$$\mathbb{E}_{x \sim \text{Unif}(D_n)} \left[ (f_{\boldsymbol{\theta}^*}(x) - y(x))) \left( f_{\boldsymbol{\theta}^*}(x) - f_{\boldsymbol{\theta}}(x) \right) \right] \leq \frac{\lambda}{2} \sqrt{\frac{1}{2^d}} \, \|\mathbf{H} \left( f_{\boldsymbol{\theta}^*}(\mathbf{X}) - f_{\boldsymbol{\theta}}(\mathbf{X}) \right)\|_1 \tag{67}$$

*Then, $\Pr(\mathcal{E}_3) \geq 1 - \delta$.*

*Proof.* The proof of this result is similar to that of Lemma 4, and we include the details for completeness. For each sample $x' \in D_n$, define the noise in the sample as $z(x') = f_{\boldsymbol{\theta}^*}(x') - y(x)$. Likewise, define the noise vector $\mathbf{z}_{D_n} = \{\mathbb{E}_{x \sim \text{Unif}(D_n)}[z(x')\mathbb{I}(x' = x)] : x \in [2^d]\}$. Then, by an application of Holder's inequality, we have,

$$\mathbb{E}_{x \sim \text{Unif}(D_n)} \left[ (f_{\boldsymbol{\theta}^*}(x) - y(x))) \left( f_{\boldsymbol{\theta}^*}(x) - f_{\hat{\boldsymbol{\theta}}}(x) \right) \right] \leq \|\mathbf{H}\mathbf{z}_{D_n}\|_\infty \, \|\mathbf{H} \left( f_{\boldsymbol{\theta}^*}(\mathbf{X}) - f_{\hat{\boldsymbol{\theta}}}(\mathbf{X}) \right)\|_1 . \tag{68}$$

Note that for each fixed row $i \in [2^d]$, $\langle \mathbf{H}_i, \mathbf{z}_{D_n} \rangle = \sum_{j \in 2^d} \mathbf{H}_{ij} \mathbf{z}_{D_n}(j)$. Note that the coordinates of $\mathbf{z}_{D_n}$ are independently distributed and subgaussian. Therefore, by Bernstein's inequality,

$$\Pr \left( \langle \mathbf{H}_i, \mathbf{z}_{D_n} \rangle \geq 3 \sqrt{\frac{\left( \sum_{j \in [2^d]} \text{Var}(\mathbf{z}_{D_n}(j)) \right) \log(1/\delta)}{2^d}} \right) \leq \delta \tag{69}$$

Note that the coordinate of $\mathbf{z}_{D_n}$ labelled by $x \in \{\pm 1\}^d$, $\mathbb{E}_{x \sim \mathrm{Unif}(D_n)}[z(x')\mathbb{I}(x' = x)]$, is the sum of $D_n(x)$ independent $\mathcal{N}(0, \sigma^2)$ Gaussians scaled by $1/n$, where $D_n(x)$, defined as the number of times $x$ is sampled in $D_n$. Therefore, $\mathrm{Var}(\mathbf{z}_{D_n}(i)) = \frac{\sigma^2 D_n(x)}{n^2}$. By union bounding over the $2^d$ rows of $\mathbf{H}$, with probability $\geq 1 - \delta$,

$$\Pr\left( \|\mathbf{H}\mathbf{z}_{D_n}\|_\infty \geq 3\sqrt{\frac{\left(\sum_{x \in [2^d]} \frac{\sigma^2 D_n(x)}{n^2}\right) \log(2^d/\delta)}{2^d}} \right) \leq \delta. \tag{70}$$

Note that $\sum_{x \in [2^d]} D_n(x) = n$, and therefore, with probability $\geq 1 - \delta$,

$$\|\mathbf{H}\mathbf{z}_{D_n}\|_\infty \leq 3\sigma\sqrt{\frac{1}{2^d}}\sqrt{\frac{d + \log(1/\delta)}{n}} \leq \frac{\lambda}{4\sqrt{2^d}}. \tag{71}$$

where the last inequality follows by the assumption on $\lambda$. This implies that with probability $\geq 1 - \delta$,

$$\mathbb{E}_{x \sim \mathrm{Unif}(D_n)}\left[ (f_{\boldsymbol{\theta}^*}(x) - y(x)))(f_{\boldsymbol{\theta}^*}(x) - f_{\hat{\boldsymbol{\theta}}}(x)) \right] \leq \frac{\lambda}{2}\sqrt{\frac{1}{2^d}}\left\| \mathbf{H}\left( f_{\boldsymbol{\theta}^*}(\mathbf{X}) - f_{\hat{\boldsymbol{\theta}}}(\mathbf{X}) \right) \right\|_1 \tag{72}$$

$\square$

**Lemma 8.** *Assume that $d \geq 3$ and $k \leq 2^d/4$. Then, there exists a set of $2^{(d-1)\lfloor k/2 \rfloor}$ binary vectors of length $2^d$, denoted $\mathcal{C}_k$, each having at most $k$ ones, such that: the hamming distance between any pair of vectors is at least $\lfloor \frac{k}{2} \rfloor$.*

*Proof.* The number of vectors within hamming distance $\kappa = \lfloor k/2 \rfloor$ of any vector is $\binom{2^d}{\kappa}$. Note that $k - \kappa \geq \kappa$.

We can greedily construct a packing of size at least $\binom{2^d}{k}/\binom{2^d}{\kappa} \geq \frac{(2^d - \kappa)!}{(2^d - k)!} \geq 2^{d(k-\kappa)}\left(\frac{3}{4}\right)^{k-\kappa} \geq 2^{d\kappa}\left(\frac{3}{4}\right)^\kappa \geq 2^{(d-1)\kappa}$ vectors before running out of binary vectors to choose. By construction, every pair of vectors has Hamming distance at least $\kappa$. $\square$

## C  Proof of Theorem 1 - Statistical performance under RSI

In this section we discuss the proof of Theorem 1. We prove a slightly more general result which characterizes the performance of stationary points of the MSE with spectral regularization under the RSI, when the regularization parameter is chosen arbitrarily.

**Theorem 4.** *Suppose the regularization parameter $\lambda$ satisfies Condition 1. Namely,*

$$\lambda \geq 12\sigma\sqrt{\frac{d + \log(1/\delta)}{n}} \tag{73}$$

*Consider a learner which returns any first order stationary point of the loss $\mathcal{L}_n(\boldsymbol{\theta}) + R(\boldsymbol{\theta})$. Under Assumptions 1(a), 2 and 3, if $\frac{C_{n,\delta}^*}{2} - \frac{3}{2}(2d + 1)\lambda\mu - 3\lambda L\sqrt{k} > 0$, with probability $\geq 1 - 2\delta$,*

$$\|\hat{\boldsymbol{\theta}} - \boldsymbol{\theta}^*\|_2 \leq \frac{6\lambda L\sqrt{k}}{C_{n,\delta}^*} \tag{74}$$

*Proof.* Plugging in $\boldsymbol{\theta} = \hat{\boldsymbol{\theta}}$ into Lemma 3, and noting that $0 \in \nabla\mathcal{L}_n(\hat{\boldsymbol{\theta}}) + (\nabla R)(\hat{\boldsymbol{\theta}})$, choosing $G$ appropriately we obtain, conditioned on $\mathcal{E}_{\mathrm{rsi}}$,

$$2\mathbb{E}_{x \sim \mathrm{Unif}(D_n)}\left[ (f_{\boldsymbol{\theta}^*}(x) - y(x)))\left\langle \boldsymbol{\theta}^* - \hat{\boldsymbol{\theta}}, \nabla f_{\hat{\boldsymbol{\theta}}}(x) \right\rangle \right]$$
$$\geq R(\hat{\boldsymbol{\theta}}) - R(\boldsymbol{\theta}^*) + \left( C_{n,\delta}^* - (2d + 1)\lambda\mu \right)\|\hat{\boldsymbol{\theta}} - \boldsymbol{\theta}^*\|_2^2 \tag{75}$$

Now, we upper bound the LHS of this equation using Lemma 4. Plugging Lemma 4 into eq. (75) with the choice of $\boldsymbol{\theta} = \hat{\boldsymbol{\theta}}$, and rearranging both sides, under the event $\mathcal{E}_{\mathrm{rsi}}$ and $\mathcal{E}_2$ which jointly occur with probability $\geq 1 - 2\delta$,

$$\left( C_{n,\delta}^* - \frac{3}{2}(2d+1)\lambda\mu \right) \|\hat{\boldsymbol{\theta}} - \boldsymbol{\theta}^*\|_2^2$$

$$\overset{(i)}{\leq} \frac{\lambda}{\sqrt{2^d}} \left\| \mathbf{H} f_{\boldsymbol{\theta}^*}(\mathbf{X}) \right\|_1 - \frac{\lambda}{\sqrt{2^d}} \left\| \mathbf{H} f_{\hat{\boldsymbol{\theta}}}(\mathbf{X}) \right\|_1 + \frac{\lambda}{2\sqrt{2^d}} \left\| \mathbf{H} \left( f_{\boldsymbol{\theta}^*}(\mathbf{X}) - f_{\hat{\boldsymbol{\theta}}}(\mathbf{X}) \right) \right\|_1 \tag{76}$$

$$\overset{(ii)}{\leq} \frac{\lambda}{\sqrt{2^d}} \left( \left\| \mathbf{H} f_{\boldsymbol{\theta}^*}(\mathbf{X}) \right\|_1 - \left\| \mathbf{H} f_{\hat{\boldsymbol{\theta}}}(\mathbf{X}) \right\|_1 + \frac{1}{2} \left\| \mathbf{H} f_{\boldsymbol{\theta}^*}(\mathbf{X}) \right\|_1 + \frac{1}{2} \left\| \mathbf{H} f_{\hat{\boldsymbol{\theta}}}(\mathbf{X}) \right\|_1 \right) \tag{77}$$

$$= \frac{\lambda}{2\sqrt{2^d}} \left( 3 \left\| \mathbf{H} f_{\boldsymbol{\theta}^*}(\mathbf{X}) \right\|_1 - \left\| \mathbf{H} f_{\hat{\boldsymbol{\theta}}}(\mathbf{X}) \right\|_1 \right) \tag{78}$$

where $(i)$ follows from the definition of the regularization term, $R(\boldsymbol{\theta}) = \frac{\lambda}{\sqrt{2^d}} \left\| \mathbf{H} f_{\boldsymbol{\theta}}(\mathbf{X}) \right\|_1$. On the other hand, $(ii)$ follows by triangle inequality of the norm $\|\cdot\|_1$. By the assumption $\frac{1}{2} C_{n,\delta}^* \geq \frac{3}{2}(2d+1)\lambda\mu + 3\lambda L\sqrt{k}$, the LHS of eq. (78) is non-negative. Plugging this into eq. (78) results in the inequality,

$$\frac{\lambda}{2\sqrt{2^d}} \left( 3 \left\| \mathbf{H} f_{\boldsymbol{\theta}^*}(\mathbf{X}) \right\|_1 - \left\| \mathbf{H} f_{\hat{\boldsymbol{\theta}}}(\mathbf{X}) \right\|_1 \right) \geq 0. \tag{79}$$

Conditioned on $\mathcal{E}_2$ and $\mathcal{E}_{\mathrm{rsi}}$.

Next, we apply (Loh & Wainwright, 2015, Lemma 5) to the function $\rho_\lambda(\cdot) = \|\cdot\|_1$, and note that by assumption $\mathbf{H} f_{\boldsymbol{\theta}^*}(\mathbf{X})$ is $k$-sparse. Define $\nu = \mathbf{H}(f_{\boldsymbol{\theta}^*}(\mathbf{X}) - f_{\hat{\boldsymbol{\theta}}}(\mathbf{X}))$ and $A$ as the set of $k$ largest indices of $\nu$ in absolute value. Using the fact that $3 \left\| \mathbf{H} f_{\boldsymbol{\theta}^*}(\mathbf{X}) \right\|_1 - \left\| \mathbf{H} f_{\hat{\boldsymbol{\theta}}}(\mathbf{X}) \right\|_1 \geq 0$,

$$\frac{\lambda}{2\sqrt{2^d}} \left( 3 \left\| \mathbf{H} f_{\boldsymbol{\theta}^*}(\mathbf{X}) \right\|_1 - \left\| \mathbf{H} f_{\hat{\boldsymbol{\theta}}}(\mathbf{X}) \right\|_1 \right) \overset{(i)}{\leq} \frac{\lambda}{\sqrt{2^d}} \left( 3\|\nu_A\|_1 - \|\nu_{A^c}\|_1 \right) \tag{80}$$

$$\leq \frac{3\lambda\sqrt{k}}{2\sqrt{2^d}} \|\nu_A\|_2 \tag{81}$$

$$\leq \frac{3\lambda\sqrt{k}}{2\sqrt{2^d}} \|\nu\|_2. \tag{82}$$

Plugging this back into eq. (78) results in the following inequality, conditioned on the event $\mathcal{E}_2$ and $\mathcal{E}_{\mathrm{rsi}}$,

$$\left( C_{n,\delta}^* - \frac{3}{2}(2d+1)\lambda\mu \right) \|\hat{\boldsymbol{\theta}} - \boldsymbol{\theta}^*\|_2^2 \leq \frac{3\lambda\sqrt{k}}{2\sqrt{2^d}} \|f_{\boldsymbol{\theta}^*}(\mathbf{X}) - f_{\hat{\boldsymbol{\theta}}}(\mathbf{X})\|_2 \tag{83}$$

Finally, we relate $\|f_{\boldsymbol{\theta}^*}(\mathbf{X}) - f_{\hat{\boldsymbol{\theta}}}(\mathbf{X})\|_2$ to $\|\hat{\boldsymbol{\theta}} - \boldsymbol{\theta}^*\|_2$.

Plugging the bound in Lemma 1 into eq. (83) results in the following inequality, conditioned on $\mathcal{E}_2$ and $\mathcal{E}_{\mathrm{rsi}}$,

$$\left( C_{n,\delta}^* - \frac{3}{2}(2d+1)\lambda\mu \right) \|\hat{\boldsymbol{\theta}} - \boldsymbol{\theta}^*\|_2^2 \leq 3\mu\lambda\sqrt{k}\|\hat{\boldsymbol{\theta}} - \boldsymbol{\theta}^*\|_2^2 + 3\lambda L\sqrt{k}\|\hat{\boldsymbol{\theta}} - \boldsymbol{\theta}^*\|_2. \tag{84}$$

Furthermore, recalling the assumption that $\frac{C_{n,\delta}^*}{2} - \frac{3}{2}(2d+1)\lambda\mu - 3\mu\lambda\sqrt{k} \geq 0$, this results in the overall bound,

$$\|\hat{\boldsymbol{\theta}} - \boldsymbol{\theta}^*\|_2 \leq \frac{3\lambda L\sqrt{k}}{C_{n,\delta}^* - \frac{3}{2}(2d+1)\lambda\mu - 3\mu\lambda\sqrt{k}} \leq \frac{6\lambda L\sqrt{k}}{C_{n,\delta}^*}, \tag{85}$$

under the events $\mathcal{E}_2$ and $\mathcal{E}_{\mathrm{rsi}}$ which jointly occur with probability $\geq 1 - 2\delta$.

$\square$

**Proof of Theorem 1.** Theorem 1 follows from Theorem 4 by choosing $\lambda$ as its lower bound in Condition 1, equal to $12\sigma\sqrt{\frac{d+\log(1/\delta)}{n}}$. It is easily verified that when the size of the dataset, $n$, is larger than the quantity $n_0$ as defined in the statement of Theorem 1, the condition $\frac{C^*_{n,\delta}}{2} - \frac{3}{2}(2d+1)\lambda\mu - 3\lambda L\sqrt{k} > 0$ as required in Theorem 4 is satisfied.

**Notes on the constants.** We did not choose to optimize the constants in the theorem statements to keep the proofs simple. With more careful analysis they can be brought down further.

## D  Proof of Theorem 2 - **Statistical performance under QG condition**

In this section we discuss the proof of Theorem 2. This result is a consequence of a more general result which characterizes the performance of stationary points of the MSE with spectral regularization when the regularization parameter, $\lambda$, is chosen arbitrarily.

**Theorem 5.** *Define* $\Delta = \frac{C^*_{n,\delta}}{2\mu} - \frac{(2d+1)\lambda}{2} - \frac{3}{2}\lambda\sqrt{k}$ *and assume* $\Delta > 0$. *Suppose the regularization parameter* $\lambda$ *satisfies Condition 1. Namely,*

$$\lambda \geq 12\sigma\sqrt{\frac{d+\log(1/\delta)}{n}} \tag{86}$$

*Consider a learner which returns a* $\Delta^2$-*approximate first order stationary interpolator (Definition 3) of the loss* $\mathcal{L}_n(\boldsymbol{\theta}) + R(\boldsymbol{\theta})$. *Then, under Assumptions 1(b), 2 and 3, with probability* $\geq 1 - 2\delta$,

$$\|\hat{\boldsymbol{\theta}} - \boldsymbol{\theta}^*\|_2 \leq \frac{3\lambda L\sqrt{k}}{C^*_{n,\delta}}. \tag{87}$$

*Proof.* Plugging in $\boldsymbol{\theta} = \hat{\boldsymbol{\theta}}$ into Lemma 6, and noting that the learner returns a first order stationary point of the regularized loss, $0 \in \nabla\mathcal{L}_n(\hat{\boldsymbol{\theta}}) + (\nabla R)(\hat{\boldsymbol{\theta}})$, choosing $G$ appropriately we have, that under the event $\mathcal{E}_{\text{qg}}$,

$$2\mathbb{E}_{x\sim\text{Unif}(D_n)}\left[\left(f_{\boldsymbol{\theta}^*}(x) - y(x)\right)\left(f_{\boldsymbol{\theta}^*}(x) - f_{\hat{\boldsymbol{\theta}}}(x)\right)\right] \tag{88}$$

$$\geq 2\text{Err}_n(\hat{\boldsymbol{\theta}}, \boldsymbol{\theta}^*) + R(\hat{\boldsymbol{\theta}}) - R(\boldsymbol{\theta}^*) - \left((2d+1)\lambda\mu + 2\mu\sqrt{\mathcal{L}_n(\hat{\boldsymbol{\theta}})}\right)\|\hat{\boldsymbol{\theta}} - \boldsymbol{\theta}^*\|_2^2 \tag{89}$$

Under assumption (A1), recall that we assume that $\hat{\boldsymbol{\theta}}$ is a sufficiently good interpolator in that, $\mathcal{L}_n(\hat{\boldsymbol{\theta}}) \leq \left(\frac{C^*_{n,\delta}}{2\mu} - \frac{(2d+1)\lambda}{2} - \frac{3}{2}\sqrt{k}\lambda\right)^2$. Simplifying eq. (89) under this assumption gives,

$$2\mathbb{E}_{x\sim\text{Unif}(D_n)}\left[\left(f_{\boldsymbol{\theta}^*}(x) - y(x)\right)\left(f_{\boldsymbol{\theta}^*}(x) - f_{\hat{\boldsymbol{\theta}}}(x)\right)\right]$$
$$\geq 2\text{Err}_n(\hat{\boldsymbol{\theta}}, \boldsymbol{\theta}^*) + R(\hat{\boldsymbol{\theta}}) - R(\boldsymbol{\theta}^*) - \left(C^*_{n,\delta} - 3\mu\lambda\sqrt{k}\right)\|\hat{\boldsymbol{\theta}} - \boldsymbol{\theta}^*\|_2^2 \tag{90}$$

Next we bound the LHS of eq. (90) using Lemma 7.

Plugging Lemma 7 into (90) and rearranging both sides, under the event $\mathcal{E}_3$,

$$2\text{Err}_n(\hat{\boldsymbol{\theta}}, \boldsymbol{\theta}^*) - \left(C^*_{n,\delta} - 3\lambda\mu\sqrt{k}\right)\|\hat{\boldsymbol{\theta}} - \boldsymbol{\theta}^*\|_2^2$$

$$\overset{(i)}{\leq} \frac{\lambda}{\sqrt{2^d}}\|\mathbf{H}f_{\boldsymbol{\theta}^*}(\mathbf{X})\|_1 - \frac{\lambda}{\sqrt{2^d}}\|\mathbf{H}f_{\hat{\boldsymbol{\theta}}}(\mathbf{X})\|_1 + \frac{\lambda}{2\sqrt{2^d}}\|\mathbf{H}\left(f_{\boldsymbol{\theta}^*}(\mathbf{X}) - f_{\hat{\boldsymbol{\theta}}}(\mathbf{X})\right)\|_1 \tag{91}$$

$$\overset{(ii)}{\leq} \frac{\lambda}{\sqrt{2^d}}\left(\|\mathbf{H}f_{\boldsymbol{\theta}^*}(\mathbf{X})\|_1 - \|\mathbf{H}f_{\hat{\boldsymbol{\theta}}}(\mathbf{X})\|_1 + \frac{1}{2}\|\mathbf{H}f_{\boldsymbol{\theta}^*}(\mathbf{X})\|_1 + \frac{1}{2}\|\mathbf{H}f_{\hat{\boldsymbol{\theta}}}(\mathbf{X})\|_1\right) \tag{92}$$

$$= \frac{\lambda}{2\sqrt{2^d}}\left(3\|\mathbf{H}f_{\boldsymbol{\theta}^*}(\mathbf{X})\|_1 - \|\mathbf{H}f_{\hat{\boldsymbol{\theta}}}(\mathbf{X})\|_1\right) \tag{93}$$

where $(i)$ follows from the definition of the regularization term, $R(\boldsymbol{\theta}) = \frac{\lambda}{\sqrt{2^d}}\|\mathbf{H}f_{\boldsymbol{\theta}}(\mathbf{X})\|_1$. On the other hand, $(ii)$ follows by triangle inequality of the norm $\|\cdot\|_1$. Next we focus on the LHS of the above expression and

simplify it further. By the quadratic growth condition in assumption 1(b), under the event $\mathcal{E}_{\mathrm{qg}} : \mathrm{Err}_n(\hat{\boldsymbol{\theta}}, \boldsymbol{\theta}^*) \geq C^*_{n,\delta}\|\hat{\boldsymbol{\theta}} - \boldsymbol{\theta}^*\|_2^2$, which is assumed to happen with probability $\geq 1 - \delta$. Therefore, under $\mathcal{E}_3$ and $\mathcal{E}_{\mathrm{qg}}$,

$$2\mathrm{Err}_n(\hat{\boldsymbol{\theta}}, \boldsymbol{\theta}^*) - \left(C^*_{n,\delta} - 3\lambda\mu\sqrt{k}\right)\|\hat{\boldsymbol{\theta}} - \boldsymbol{\theta}^*\|_2^2 \tag{94}$$

$$\geq \|\hat{\boldsymbol{\theta}} - \boldsymbol{\theta}^*\|_2^2\left(C^*_{n,\delta} + 3\lambda\mu\sqrt{k}\right) \tag{95}$$

$$\geq 0. \tag{96}$$

Plugging this into eq. (93) results in the inequality,

$$\frac{\lambda}{2\sqrt{2^d}}\left(3\left\|\mathbf{H}f_{\boldsymbol{\theta}^*}(\mathbf{X})\right\|_1 - \left\|\mathbf{H}f_{\hat{\boldsymbol{\theta}}}(\mathbf{X})\right\|_1\right) \geq 0. \tag{97}$$

Under the events $\mathcal{E}_3$ and $\mathcal{E}_{\mathrm{qg}}$.

Next, we apply (Loh & Wainwright, 2015, Lemma 5) to the function $\rho_\lambda(\cdot) = \|\cdot\|_1$, and note that by assumption $\mathbf{H}f_{\boldsymbol{\theta}^*}(\mathbf{X})$ is $k$-sparse. By defining $\nu = \mathbf{H}(f_{\boldsymbol{\theta}^*}(\mathbf{X}) - f_{\hat{\boldsymbol{\theta}}}(\mathbf{X}))$ and $A$ as the set of $k$ largest indices of $\nu$ in absolute value.

$$\frac{\lambda}{2\sqrt{2^d}}\left(3\left\|\mathbf{H}f_{\boldsymbol{\theta}^*}(\mathbf{X})\right\|_1 - \left\|\mathbf{H}f_{\hat{\boldsymbol{\theta}}}(\mathbf{X})\right\|_1\right) \overset{(i)}{\leq} \frac{\lambda}{\sqrt{2^d}}\left(3\|\nu_A\|_1 - \|\nu_{A^c}\|_1\right) \tag{98}$$

$$\leq \frac{3\lambda\sqrt{k}}{2\sqrt{2^d}}\|\nu_A\|_2 \tag{99}$$

$$\leq \frac{3\lambda\sqrt{k}}{2\sqrt{2^d}}\|\nu\|_2. \tag{100}$$

here, $(i)$ uses the fact that $3\left\|\mathbf{H}f_{\boldsymbol{\theta}^*}(\mathbf{X})\right\|_1 - \left\|\mathbf{H}f_{\hat{\boldsymbol{\theta}}}(\mathbf{X})\right\|_1 \geq 0$ from eq. (97).

Finally, we plug the relation between $\|v\|_2 = \|f_{\boldsymbol{\theta}^*}(\mathbf{X}) - f_{\hat{\boldsymbol{\theta}}}(\mathbf{X})\|_2$ to $\|\hat{\boldsymbol{\theta}} - \boldsymbol{\theta}^*\|_2$ proved in Lemma 1 into eq. (100). This results in the inequality,

$$\frac{\lambda}{2\sqrt{2^d}}\left(3\left\|\mathbf{H}f_{\boldsymbol{\theta}^*}(\mathbf{X})\right\|_1 - \left\|\mathbf{H}f_{\hat{\boldsymbol{\theta}}}(\mathbf{X})\right\|_1\right) \leq 3\mu\lambda\sqrt{k}\|\hat{\boldsymbol{\theta}} - \boldsymbol{\theta}^*\|_2^2 + 3\lambda L\sqrt{k}\|\hat{\boldsymbol{\theta}} - \boldsymbol{\theta}^*\|_2 \tag{101}$$

Plugging this back into eq. (93), under $\mathcal{E}_3$ and $\mathcal{E}_{\mathrm{qg}}$,

$$2\mathrm{Err}_n(\hat{\boldsymbol{\theta}}, \boldsymbol{\theta}^*) - \left(C^*_{n,\delta} - 3\lambda\mu\sqrt{k}\right)\|\hat{\boldsymbol{\theta}} - \boldsymbol{\theta}^*\|_2^2 \leq 3\mu\lambda\sqrt{k}\|\hat{\boldsymbol{\theta}} - \boldsymbol{\theta}^*\|_2^2 + 3\lambda L\sqrt{k}\|\hat{\boldsymbol{\theta}} - \boldsymbol{\theta}^*\|_2. \tag{102}$$

Resulting in the bound,

$$2\mathrm{Err}_n(\hat{\boldsymbol{\theta}}, \boldsymbol{\theta}^*) \leq 3\lambda L\sqrt{k}\|\hat{\boldsymbol{\theta}} - \boldsymbol{\theta}^*\|_2 + C^*_{n,\delta}\|\hat{\boldsymbol{\theta}} - \boldsymbol{\theta}^*\|_2^2. \tag{103}$$

Under the quadratic growth condition, by the event $\mathcal{E}_3$ in Assumption 1(b), $\mathrm{Err}_n(\hat{\boldsymbol{\theta}}, \boldsymbol{\theta}^*) \geq C^*_{n,\delta}\|\hat{\boldsymbol{\theta}} - \boldsymbol{\theta}^*\|_2^2$. Therefore, under the events $\mathcal{E}_3$ and $\mathcal{E}_{\mathrm{qg}}$ which jointly occur with probability $\geq 1 - 2\delta$,

$$\|\hat{\boldsymbol{\theta}} - \boldsymbol{\theta}^*\|_2 \leq \frac{3\lambda L\sqrt{k}}{C^*_{n,\delta}}. \tag{104}$$

**Proof of Theorem 2.** Theorem 2 follows from Theorem 5 by choosing $\lambda$ as its lower bound in Condition 1, equal to $12\sigma\sqrt{\frac{d + \log(1/\delta)}{n}}$. It is easily verified that when $n > n_0$, as defined in the statement of Theorem 2, the condition $\Delta = \frac{C^*_{n,\delta}}{2} - \frac{1}{2}(2d + 1)\lambda\mu - \frac{3}{2}\lambda L\sqrt{k} > 0$ as required in Theorem 5 is satisfied. $\qquad\square$

# E Proof of Theorem 3

In this section, we prove a lower bound on the statistical error of parameter estimation. Loosely speaking, the objective is to show that for every learner $\hat{\boldsymbol{\theta}}$,

$$\sup_{\boldsymbol{\theta}^*} \mathbb{E}\left[\|\hat{\boldsymbol{\theta}} - \boldsymbol{\theta}^*\|_2\right] \gtrsim \sigma\sqrt{\frac{kd}{n}}. \tag{105}$$

*Proof.* We first introduce an auxiliary result related to packing binary vectors with bounded Hamming weight in Lemma 8. Now define the function space $\mathcal{G} = \{f_{\boldsymbol{\theta}}(\cdot) : \boldsymbol{\theta} \in \mathbb{R}^{2^d}\}$, where $f_{\boldsymbol{\theta}}(x)$ is defined as the polynomial with coefficients specified by $\boldsymbol{\theta}$. Namely, $f_{\boldsymbol{\theta}}(x) = \sum_{S \subseteq [d]} \theta_S \prod_{i \in S} x_i$, where we index the $2^d$ coefficients of $\theta$ by the $2^d$ subsets of $[d]$. In an alternate notation, we may represent,

$$f_{\boldsymbol{\theta}}(x) = \left\langle \theta, 2^{[x]} \right\rangle \tag{106}$$

where $2^{[x]}$ denotes the $2^d$ length vector whose element indexed by some subset $S \subseteq [d]$ is $\prod_{i \in S} x_i$.

Furthermore, we assume that the data generating distribution independently samples $n$ pairs $(x_i, y_i)$ where $x_i \sim \mathrm{Unif}(\{\pm 1\}^d)$ and $y_i = f_{\boldsymbol{\theta}^*}(x_i) + Z_i$ where $Z_i \sim \mathcal{N}(0, \sigma^2)$. Denote $D_n = \{x_1, \cdots, x_n\}$.

By Lemma 8, the binary vectors belonging to $\mathcal{C}_k$ can be used to construct a subset of the function space $\mathcal{G}$, defined as $\mathcal{S}_k$,

$$\mathcal{S}_k = \{f_{\boldsymbol{\theta}}(\cdot) : \boldsymbol{\theta} \in \Delta\mathcal{C}_k\}, \tag{107}$$

where $\Delta > 0$ is a scaling factor and $\Delta\mathcal{C}_k = \{\Delta\boldsymbol{\theta} : \boldsymbol{\theta} \in \mathcal{C}_k\}$.

Henceforth, we will consider ourselves with learning functions (resp. parameters) in the class $\mathcal{S}_k$ (resp. $\Delta\mathcal{C}_k$).

First we show the properties on $\mathrm{Err}_n(\boldsymbol{\theta}, \boldsymbol{\theta}^*)$ and $\mathcal{G}$ in the statement of Theorem 3. Note that $\mathcal{G}$ is a linear family by the representation in eq. (106), and therefore $\mu = 0$.

In addition, note that,

$$\mathrm{Err}_n(\boldsymbol{\theta}, \boldsymbol{\theta}^*) = \mathbb{E}_{x \sim D_n}\left[(f_{\boldsymbol{\theta}}(x) - f_{\boldsymbol{\theta}^*}(x))^2\right] \tag{108}$$

$$= \mathbb{E}_{x \sim D_n}\left[\left\langle \boldsymbol{\theta} - \boldsymbol{\theta}^*, 2^{[x]} \right\rangle^2\right] \tag{109}$$

$$= (\boldsymbol{\theta} - \boldsymbol{\theta}^*)^T \mathbb{E}_{x \sim D_n}\left[2^{[x]}(2^{[x]})^T\right](\boldsymbol{\theta} - \boldsymbol{\theta}^*) \tag{110}$$

Note that $A_x = 2^{[x]}(2^{[x]})^T$ is a matrix whose entries (indexed by pairs of subsets of $[d]$) can be described as $A_x(S, T) = \prod_{i \in S} x_i \prod_{j \in S} x_j$. Note that in expectation over $x \sim \mathrm{Unif}(\mathbf{X})$, we have that,

$$\mathbb{E}_{x \sim \mathrm{Unif}(\mathbf{X})}[A_x(S, T)] = \mathbb{I}(S = T) \tag{111}$$

This is because if $S \neq T$, there exists an element $i \in (S \setminus T) \cup (T \setminus S)$ (i.e. the symmetric difference of the two sets) and since $\mathbb{E}_{x \sim \mathrm{Unif}(\mathbf{X})}[x_i] = 0$, we get the required statement. Therefore,

$$\mathbb{E}_{x \sim \mathrm{Unif}(\mathbf{X})}[A_x] = I \tag{112}$$

Now, given $n$ samples from the uniform distribution, $\mathbb{E}_{x \sim \mathrm{Unif}(D_n)}[A_x]$ is expected to concentrate around its expectation $\mathbb{E}_{x \sim \mathrm{Unif}(\mathbf{X})}[A_x]$. In particular, by invoking the matrix Bernstein inequality Tropp et al. (2015), we have that,

$$\Pr\left(\left\|\mathbb{E}_{x \sim \mathrm{Unif}(D_n)}[A_x] - \mathbb{E}_{x \sim \mathrm{Unif}(\mathbf{X})}[A_x]\right\|_{\mathrm{op}} \geq t\right) \leq 2(2^d)\exp\left(-nt^2/2L^2\right) \tag{113}$$

where $L$ is an almost sure upper bound on $\|A_x\|_{\mathrm{op}}$. Note that $\|A_x\|_{\mathrm{op}} = \|2^{[x]}\|_2 = \sqrt{2^d}$ and therefore we may choose $L = \sqrt{2^d}$. This results in the bound,

$$\Pr\left(\left\|\mathbb{E}_{x\sim\mathrm{Unif}(D_n)}[A_x] - I\right\|_{\mathrm{op}} \geq \frac{1}{2}\right) \leq 2(2^d)\exp\left(-n/2^{d+3}\right) \tag{114}$$

Therefore, if $n \gtrsim (d + \log(1/\delta))2^{d+3}$, with probability $\geq 1 - \delta$,

$$\frac{1}{2}I \preceq \mathbb{E}_{x\sim\mathrm{Unif}(D_n)}[A_x] \preceq \frac{3}{2}I. \tag{115}$$

In eq. (110), this implies that, with probability $\geq 1 - \delta$,

$$\mathrm{Err}_n(\boldsymbol{\theta}, \boldsymbol{\theta}^*) \geq \frac{1}{2}\|\boldsymbol{\theta} - \boldsymbol{\theta}^*\|_2^2 \tag{116}$$

And likewise, from the linear representation in eq. (106),

$$\mathbb{E}_{x\sim\mathrm{Unif}(\mathbf{X})}\left[\nabla f_{\boldsymbol{\theta}^*}(x)(\nabla f_{\boldsymbol{\theta}^*}(x))^T\right] = \mathbb{E}_{x\sim\mathrm{Unif}(\mathbf{X})}\left[2^{[x]}(2^{[x]})^T\right] = I \tag{117}$$

where the last equation follows from eq. (112).

To generate the lower bound instance, suppose the ground truth parameter $\boldsymbol{\theta}^*$ is sampled uniformly from $\Delta\mathcal{C}_k$. Suppose the learner outputs a candidate parameter $\hat{\boldsymbol{\theta}}$. The population level mean squared error of the learner is lower bounded by the testing error,

$$\sup_{\boldsymbol{\theta}^*\in\mathbb{R}^{2^d}}\mathbb{E}\left[\|\hat{\boldsymbol{\theta}} - \boldsymbol{\theta}^*\|_2\right] \geq \mathbb{E}_{\boldsymbol{\theta}^*\sim\mathrm{Unif}(\Delta\mathcal{C}_k)}\left[\mathbb{E}\left[\left\|\hat{\boldsymbol{\theta}} - \boldsymbol{\theta}^*\right\|_2\right]\right] \tag{118}$$

$$\geq \frac{1}{4}\sqrt{\lfloor k/2\rfloor}\Delta\inf_{\Psi}\mathbb{E}_{\boldsymbol{\theta}^*\sim\mathrm{Unif}(\mathcal{C}_k)}\left[\mathbb{I}(\Psi(D_n)) \neq \boldsymbol{\theta}^*)\right] \tag{119}$$

where the infimum is over all tests functions which return a function in $\mathcal{C}_k$. This uses the fact that any estimator $\hat{\boldsymbol{\theta}}$ induces a testing function for $\boldsymbol{\theta}^*$ by returning the $\boldsymbol{\theta} \in \Delta\mathcal{C}_k$ such that $\|\boldsymbol{\theta} - \hat{\boldsymbol{\theta}}\|_2$ is smallest. If this test makes a mistake, then $\hat{\boldsymbol{\theta}}$ must have predicted $> \frac{1}{2}\sqrt{\lfloor k/2\rfloor}$ entries of $\boldsymbol{\theta}^*$ as $< \Delta/2$ instead of $\Delta$ or $> \Delta/2$ instead of $0$ (making an error of at least $\Delta/2$ on these coordinates).

The optimal hypothesis testing error can be lower bounded by Fano's inequality as follows,

$$\inf_{\Psi}\mathbb{E}_{\boldsymbol{\theta}^*\sim\mathrm{Unif}(\Delta\mathcal{C}_k)}\left[\mathbb{I}(\Psi(D_n)) \neq \boldsymbol{\theta}^*)\right] \geq 1 - \frac{I(\boldsymbol{\theta}^*; D_n) + \log(2)}{\log|\Delta\mathcal{C}_k|} \tag{120}$$

For each parameter $\boldsymbol{\theta} \in \Delta\mathcal{C}_k$, define the distribution,

$$P_{\boldsymbol{\theta}}(D_n) = \prod_{i=1}^{n}\frac{1}{\sqrt{2\pi}}\exp\left(-\frac{1}{2\sigma^2}(y_i - f_{\boldsymbol{\theta}}(x_i))^2\right) \times \left(\frac{1}{2^d}\right). \tag{121}$$

which captures the data distribution when the underlying ground truth parameter is $\boldsymbol{\theta}$. Note that $D_n \sim P_{\boldsymbol{\theta}^*}$; marginally $x_i$ is uniformly distributed on the hypercube and conditionally $y_i$ is normally distributed with mean $f_{\boldsymbol{\theta}^*}(x_i)$ and variance $\sigma^2$. Note that the mutual information can be bounded as,

$$I(\boldsymbol{\theta}^*, D_n) = \inf_{Q}\mathbb{E}_{\boldsymbol{\theta}^*\sim\mathrm{Unif}(\Delta\mathcal{C}_k)}\left[\mathsf{KL}(P_{\boldsymbol{\theta}^*}\|Q)\right] \tag{122}$$

$$\leq \mathbb{E}_{\boldsymbol{\theta}^*\sim\mathrm{Unif}(\Delta\mathcal{C}_k)}\left[\mathsf{KL}(P_{\boldsymbol{\theta}^*}\|P_0)\right] \tag{123}$$

where $P_0$ is the distribution of $D_n$ when the ground truth function $f_{\boldsymbol{\theta}}$ in eq. (121) is chosen as $0$ everywhere. Then,

$$\mathsf{KL}(P_{\boldsymbol{\theta}^*}\|P_0) = \frac{1}{2\sigma^2}\mathbb{E}_{D_n\sim P_{\boldsymbol{\theta}^*}}\left[\sum_{i=1}^{n}2y_if_{\boldsymbol{\theta}^*}(x_i) - (g_{\boldsymbol{\theta}}^*(x_i))^2\right] \tag{124}$$

$$= \frac{1}{2\sigma^2}\mathbb{E}_{D_n\sim P_{\boldsymbol{\theta}^*}}\left[\sum_{i=1}^{n}(f_{\boldsymbol{\theta}^*}(x_i))^2\right] \tag{125}$$

$$= \frac{n}{2\sigma^2}\mathbb{E}_{x_i\sim\mathrm{Unif}(\{\pm1\}^d)}\left[(f_{\boldsymbol{\theta}^*}(x_i))^2\right] \tag{126}$$

where the last equation just uses the fact that in $P_{\boldsymbol{\theta}^*}$, the marginal distribution of $x_i$ is uniform. For each $\boldsymbol{\theta}^*$, by Parseval's theorem, $\mathbb{E}_{x_i \sim \mathrm{Unif}(\{\pm 1\}^d)}\left[(f_{\boldsymbol{\theta}^*}(x_i))^2\right] = k\Delta^2$. Overall, plugging into eq. (126) and subsequently into eq. (123),

$$I(\boldsymbol{\theta}^*, D_n) \leq \mathbb{E}_{\boldsymbol{\theta}^* \sim \mathrm{Unif}(\Delta \mathcal{C}_k)}\left[\mathsf{KL}(P_{\boldsymbol{\theta}^*} \| P_0)\right] = \frac{nk\Delta^2}{2\sigma^2} \tag{127}$$

Putting these bounds into eq. (120) and subsequently into eq. (119) results in,

$$\mathbb{E}_{\boldsymbol{\theta}^* \sim \mathrm{Unif}(\Delta \mathcal{C}_k)}\left[\mathbb{E}\left[\left\|\hat{\boldsymbol{\theta}} - \boldsymbol{\theta}^*\right\|_2\right]\right] \geq \frac{1}{4}\sqrt{\lfloor k/2 \rfloor}\Delta\left(1 - \frac{nk\Delta^2/2\sigma^2 + \log(2)}{\log 2^{(d-1)\lfloor k/2 \rfloor}}\right) \tag{128}$$

$$\geq \frac{1}{4}\sqrt{\lfloor k/2 \rfloor}\Delta\left(1 - 10\frac{(nk\Delta^2/\sigma^2 + 1)}{(d-1)\lfloor k/2 \rfloor}\right) \tag{129}$$

Choosing $\Delta = 8\epsilon/\sqrt{\lfloor k/2 \rfloor}$, for a sufficiently large constant $C$, we get that for any learner, if $n \leq C\frac{\sigma^2 d}{\Delta^2} \asymp \frac{\sigma^2 kd}{\epsilon^2}$,

$$\mathbb{E}_{\boldsymbol{\theta}^* \sim \mathrm{Unif}(\Delta \mathcal{C}_k)}\left[\mathbb{E}\left[\left\|\hat{\boldsymbol{\theta}} - \boldsymbol{\theta}^*\right\|_2\right]\right] \geq \epsilon \asymp \sigma\sqrt{\frac{kd}{n}}. \tag{130}$$

$\square$

