# OpenReview forum: "Spectral Regularization Allows Data-frugal Learning over Combinatorial Spaces"
_TMLR — Accepted by TMLR_

### Review · Reviewer_EH8r · 2022-11-16

**Summary Of Contributions:**

The paper proposes a theoretical analysis of a Lasso-like type of regularization aiming at finding sparse representations of pseudo-boolean functions learned through empirical risk minimization. This so called spectral regularization is computed via the Walsh-Hadamard transform, and coincides in the linear model setting with the classical Lasso regularization term.

The statistical analysis of the resulting estimator is mainly divided in two parts corresponding to two different assumptions on the behavior of the MSE of the learned function compared to the ground truth, denoted by $\mathrm{Err}_n(\theta, \theta^\star)$, in addition to regularity assumptions on the model class $\mathcal{F}$ (namely Lipschitz continuous gradients of the evaluations w.r.t. the parameter, and bounded average gradient norm at $\theta^\star$).

- In the first part of the analysis, $\mathrm{Err}_n(\theta, \theta^\star)$ is assumed to satisfy a restricted secant inequality (RSI, extended notion of convexity), and Theorem 1 ensures that with high probability the estimated parameter converges in 2-norm to the ground truth parameter in the class. The proposed convergence rate scales as $\mathcal{O}(\frac{1}{\sqrt{n}})$ in terms of the number of samples, with constants involving information about the problem.

- In the second part, the RSI assumption is traded against a quadratic growth condition (QGC) on $\mathrm{Err}_n(\theta, \theta^\star)$, which does not inform about the correlation between the gradients of $\mathrm{Err}_n(\theta, \theta^\star)$ and $\theta - \theta^\star$ anymore. To alleviate this difficulty the concept of first-order interpolators is introduced, and under these new assumptions Theorem 2 proposes again a convergence rate in high probability scaling as $\mathcal{O}(\frac{1}{\sqrt{n}})$.

To complete this analysis, a lower bound result under the aforementioned assumptions is derived, showing statistical optimality with respect to the $\mathcal{O}(\frac{1}{\sqrt{n}})$ scaling.

A coherent empirical study is proposed, illustrating the interest of the spectral regularization scheme on both simulated and real data. The study goes beyond showing improvements in generalization accuracy, by exploring numerically the assumptions used in the theoretical derivations.

**Audience:**

Yes

**Broader Impact Concerns:**

Does not apply here.

**Claims And Evidence:**

Yes

**Requested Changes:**

Below are a few suggestions for adjustments, decreasingly ordered in terms of criticality.

- Plots in Figure 2 and 7 need to be extracted in better quality. The labels are barely readable.

- In assumption 2, it may be of interest to introduce the constant $\mu$ at the beginning of the sentence, as this constant should exist independently of all $x \in \mathbf{X}$. Here this does not cause real issues as $\mathbf{X}$ is discrete, but it would be cleaner.

- The proofs section organisation could be improved. Section B is supposed to be dedicated to proving Theorem 2, yet section B.1 presents the proof of Theorem 3. A suggestion would be: B) Technical lemmas C) Proof of Theorem 1 (through Theorem 5) D) Proof of Theorem 2 (through Theorem 4) E) Proof of Theorem 3. Generally speaking, it is better to not refer to some unannonced Lemma presented further down in another section as it makes the reading difficult. Also, for completion, the proof of Lemma 8 is missing.

- The proofs often suggest to "plug something into Lemma Y", it would be better if the equations were refered to instead of the lemmas. Example: the LHS of Lemma 6 is Eq (97). Proposition 1 could also be made a lemma to unify the environments used.

- I would have prefered having access to the related work section higher in the paper, between sections 2 and 3 - although I do not know how much it speaks of my personal tastes rather than some objective efficiency.

**Strengths And Weaknesses:**

Here are the strengths and weaknesses that I assessed; I added a paragraph with additional remarks that I have for the authors at the end.

Strengths:

- The presentation of the paper is good, the research questions are clearly stated in the introduction as well as in the experiments section.
- The mathematical details are sound.
- The theoretical contributions shed light on an open problem: generalization properties of the spectral regularization scheme for learning pseudo-boolean functions.
- The theoretical assumptions are not only there to provide some bounds, they are numerically checked, especially the critical quantity $C_{n,\delta}^\star$. Related to this, the discussion in the paragraph "Extensions to cases with multiple local minima" is invaluable and shows that a special care was given to this aspect of the work (having realistic assumptions).
- The experiments section is convincing.
- Some python code is provided.

Weaknesses:
- The influence of some of the quantities in the bounds is not discussed, especially $C_0$ and $C_1$ in Theorem 1. I could not understand where the $C_1$ quantity was coming from in the proofs.
- The guarantees hold for a specific path of regularization parameter $\lambda$ that depends on the (unknown) level of noise in the statistical model. This is classical in Lasso-related papers, but could have been discussed.

Additional remarks:

- QG is presented as weaker than RSI, and while I agree that this holds true for functions with Lipschitz continuous gradients, given that the loss function used is the square loss, one can say that the Lipschitz constant of the gradient of $\mathrm{Err}_n(\theta, \theta^\star)$ scales as the Lipschitz constant of the overall neural network. Lipschitz constants associated to neural networks are known to explode quite fast, so that in practice even if a function is RSI(C), the quantity 2C/L might be very small. Could the authors comment on that ?

---

> ### Author Response · Authors · 2022-12-06
> **Notes on Constants C0 and C1 and Regularization Parameter**
>
>
> We thank the reviewer for their time, valuable insights, and encouraging notes. We have detailed our response to their comments and updated the draft accordingly.
>
> > The influence of some of the quantities in the bounds is not discussed, especially $C_0$ and $C_1$ in Theorem 1. I could not understand where the $C_1$ quantity was coming from in the proofs.
>
> We did not make the constants $C_0$ and $C_1$ explicit in the proof to avoid making the theorem too messy to read. A crude bound shows that $C_0$ and $C_1$ can be chosen as $10$  . With more careful analysis it can be perhaps brought down further.
>
> Appearance of $C_1$ in the bound can be understood from Eq (72). Recall that two conditions must be satisfied by the choice of $\lambda$ in the condition of **Theorem 5**.
>
> (i) $\frac{C^*_{n,\delta}}{2} - \frac{3}{2} (2d+1) \lambda \mu - 3 \lambda L \sqrt{k} > 0$
>
> (ii) $\lambda \ge 4C_0 \sigma \sqrt{\frac{d + \log(1/\delta)}{n}}$
>
> The lower bound $n_0$ can be obtained by choosing $\lambda$ as $4C_0 \sqrt{\frac{d + \log(1/\delta)}{n}}$ and choosing $n$ to be sufficiently large that the first condition is satisfied. The leading constant that appears here in the lower bound on $n$ is what is defined as $C_1$.
>
> We have made these points clear in the revised manuscript.
>
> > The guarantees hold for a specific path of regularization parameter $\lambda$ that depends on the (unknown) level of noise in the statistical model. This is classical in Lasso-related papers, but could have been discussed.
>
> The point about requiring to choose $\lambda$ based on the knowledge of the noise parameter $\sigma^*$ (in this section we denote it $\sigma^*$ to make it clear that it is the true noise level) can be resolved in practice by cross-validation. In theory, this can be relaxed by knowing an upper bound $\sigma_{ub}$ on $\sigma^*$ known, then $\lambda$ can be chosen using this value of $\sigma_{ub}$ instead. The statements of the theorem remain unchanged, except everywhere $\sigma$ is replaced by $\sigma_{ub}$.
>
> In the linear case (and generalizations), there is another strategy to tune the regularization parameter, which is much more lenient and does not require a tight lower/upper bound on $\sigma$. This is a little more involved, but is essentially a version of cross validation, which uses the fact that the training loss of the least square estimator trained on the validation set concentrates around $\sigma^*$.
>
> Method: Pick the first n/2 points in the overall dataset as the validation set. Starting with $\sigma = \sigma_0$, assumed to be smaller than the true value $\sigma^*$ (even exponentially smaller is fine), recursively double $\sigma_t$ until the following condition is violated by $\theta = \theta_t$:
>
>  $E_{(x,y) \sim \textrm{validation set}} [ ( f_{\theta} (x) - y ) ] > \sigma_t^2 - \Delta$ (A)
>
> Here $\hat{\theta}_t$ is the solution obtained by assuming that the noise parameter is $\sigma_t$ used to define the regularization appropriately as $4 C_0 \sigma_t  \sqrt{\frac{d + \log(1/\delta)}{n}}$ and running gradient descent. Likewise, $\Delta$ is half the width of the confidence interval of the training loss of the least squares estimator.
>
> We show that the number of restarts required, $T$, is approximately $\log (\sigma^*/\sigma_0)$.
>
> The intuition is as follows: the generalization error of the least square estimator on the validation set (which is the minimizer of the LHS of eq (A)) has loss which concentrates around $(\sigma^*)^2$. This means that the condition (A) will certainly be satisfied until $\sigma_t$ grows to become larger than $\sigma^*$. On the other hand, since the algorithm guarantees good generalization (from **Theorem 1**) as long as the parameter $\sigma_t > \sigma^*$. In the first iteration $T$ when the condition $\hat{\theta}_T \ge 2 \sigma^*$ is satisfied, we have,
>
> $E_{(x,y) \sim \textrm{validation set}} [ ( f_{\hat{\theta}_T} (x) - y ) ] \overset{(i)}{\le} (\sigma^*)^2 + O( (\sigma^*)^2 \frac{k (d + \log(1/\delta))}{n} ) \le 2 (\sigma^*)^2 < (\sigma_t)^2 - \Delta$ (B)
>
> And the condition (A) is violated. Here (i) assumes that $n$ is larger than $k(d + \log (1/\delta))$ which is satisfied under the statement of **Theorem 1** by the condition $n>n_0$. This means that we have an estimate for $\sigma^*$ which is accurate up to constant multiplicative factors. Moreover, the number of rounds of hyperparameter tuning required only scales logarithmically in the initial multiplicative gap between $\sigma_0$ and $\sigma^*$.
>
> This approach can be applied in any setting as long as the the optimal training loss can be shown to admit some concentration property around $\sigma^*$, which allows the optimal training loss to be an estimator for $\sigma^*$ up to a constant multiplicative factor.

---

### Review · Reviewer_hciS · 2023-02-02

**Summary Of Contributions:**

This paper considers the problem of learning pseudo-Boolean functions via spectral regularization. While most studies on this topic consider algorithm design and applications, the paper considers theoretical analysis to show the accuracy gain of spectral regularization in learning pseudo-Boolean functions. Under a restricted secant inequality on the empirical error, the paper shows that stationary points approximate the global model as measured by the Euclidean distance. The paper further extends this result to an approximate first-order stationary interpolator under a weaker quadratic growth condition. Experimental results are also given to verify the effectiveness of spectral regularization and the theoretical statements.

**Audience:**

Yes

**Broader Impact Concerns:**

I do not see any concern on broader impacts

**Claims And Evidence:**

Yes

**Requested Changes:**

- I would request the authors to give more discussion on the restricted secant inequality and QG condition to justify these assumptions in the setting of learning pseudo-Boolean functions.

- Definition 1: it seems that $\langle \nabla g(z),z^*-z\rangle$ should be $\langle \nabla g(z),z-z^*\rangle$

- The authors should use *\\citep* instead of *\\citet* when the reference is not a part of the sentence

- Section 1: "a new generalization bounds"

- Section 1: "demonstrations on ... complements"

- Below Thm 3: "a permutations"

- End of section 4: "is also be"

- In the proof of Lemma 1 and Lemma 6, the authors use $g_\theta$. It seems that they should be $f_\theta$

- Eq (19): $2\mu$ should be $2\mu^2$

- Eq (20): $\|\hat{\theta}  -\theta^\star\|_2$ should be    $\| \hat{\theta} - \theta^\star\|_2^2$

- Eq (25): $C^*_{n,\delta}/2$ should be $C^*_{n,\delta}$

- Below eq (57): "Plugging this in,"

- It would be nice if the authors can give a reference for Proposition 1

- Eq (107): "." should be ","

**Strengths And Weaknesses:**

**Strength**

- The theoretical analysis shows that the stationary point of the regularized objective is closed to the groundtruth. An interesting property is that the upper bound depends on the sparsity parameter $k$, which shows the effect of spectral regularization to capture sparsity.

- Lower bounds are also given, which match the upper bounds and therefore show the optimality of the analysis.

- The paper is well written. The analysis seems to be rigorous. The experimental analysis is comprehensive.

**Weakness**

- Both the restricted secant inequality and the quadratic growth (QG) condition are stated for the reference point $\theta^*$, which is the groundtruth. It seems that these assumptions are stronger than the assumptions in the literature. For example, in [1], the QG assumption says $f(x)-\inf_xf(x)\geq \mu\|x-x_p\|^2$ , where $x_p$ is the projection of $x$ to the set of optimal solutions. That is, the QG assumption in [1] requires the distance to the optimal solution set is bounded by the excess function value. This paper instead considers the distance to $\theta^*$, which is a stronger assumption. Furthermore, the paper does not give a specific example which satisfies these assumptions.

- $n_0$  in Thm 1 scales as $(d^3+dk)/C_{n,\delta}^*$. This assumption can be restrictive in high-dimensional settings. This is especially the case if $C_{n,\delta}^*$ is small.

- In the proof of Theorem 3, the authors assume $n\geq d2^{d+3}$. It seems this is a restrictive assumption is $d$ is moderate.

[1]  Karimi, Hamed, Julie Nutini, and Mark Schmidt. "Linear convergence of gradient and proximal-gradient methods under the polyak-lojasiewicz condition." ECML 2016

---

> ### Author Response · Authors · 2023-02-24
> **RSI and QG conditions, n_0 scaling, and Theorem 3**
>
> > Both the Restricted Secant Inequality (RSI) and the Quadratic Growth (QG) conditions are stated for the reference point $\theta^*$, which is the groundtruth. It seems that these assumptions are stronger than the assumptions in the literature. For example, in [1], the QG assumption says $f(x) - \inf_x f(x) \geq \mu |x - x_p|^2$, where $x_p$ is the projection of $x$ to the set of optimal solutions. That is, the QG assumption in [1] requires the distance to the optimal solution set is bounded by the excess function value. This paper instead considers the distance to $\theta^*$, which is a stronger assumption. Furthermore, the paper does not give a specific example which satisfies these assumptions.
>
> As stated in [1], the RSI and QG conditions can be extended to the current paper. Indeed, the discussion in the subsection “Extensions to cases with multiple local minima” resembles the QG condition in [1]. We have rewritten this section to be more transparent in its connection to the QG condition in [1]. These assumptions are satisfied when the function class is linear in the parameter. Furthermore, the goal of the experiments in Fig 2 (and in the appendix) are indeed sanity checks to verify that these assumptions are not unreasonable for shallow neural networks.
>
> > $n_0$ in Thm. 1 scales as $(d^3 + dk) /C^*_{n,\delta}$. This assumption can be restrictive in high-dimensional settings. This is especially the case if $C^*_{n,\delta}$ is small.
>
> The bound on $n_0$ in the quadratic growth condition indeed scales as $d^3 + dk$ when $d$ is large. However, when the function classes are more restricted, in that all function $f_{\theta} \in \mathcal{F}$ have at most $O((2^d)/d)$ non-zero Hadamard coefficients, we believe that the $O(d^3)$ cost can be removed. This cost results from a bound on $\| \mathbf{H} z \|_1$ where $z \in \lbrace \pm 1 \rbrace^{2^d}$ is a binary vector, which is used in Lemma 5 (specifically eq. (52)).
>
> If $f_{\theta}$ was sparse in the spectral domain, this maximization is now over $z \in  \lbrace -1,0,+1 \rbrace^{2^d}$, under the constraint $z_i = 0$ for all $i > 2^d /d$. We conjecture that for such vectors, $\| \mathbf{H} z \|_1$ is upper bounded by $\sqrt{2^d}$, improving over the bound of $(2d+1)\sqrt{2^d}$ for unconstrained $z$. This improved bound on $\| \mathbf{H} z \|_1$ can be directly used in Lemma 5, and following through with the rest of the proof results in a bound on $n_0$ which no longer has the $O(d^3)$ cost and scales asymptotically as $O( \sigma^2 (d + \log(1/\delta) (k L^2 + \mu^2) / ({C^\ast}_\{n,\delta\})^2 )$, which scales asymptotically as $O(dk)$, and is optimal.
>
> If this conjecture were true, then the prohibitive $O(d^3)$ cost appears only when there are functions in $\mathcal{F}$ which were very dense in the spectral domain. In practice, several papers have however shown that neural networks are biased toward learning (and expressing) low-degree pseudo-boolean functions, which further reinforces why this prohibitive $O(d^3)$ cost is never observed in practice.
>
>
> > In the proof of Theorem 3, the authors assume n $\geq$ $d2^{d+3}$. It seems this is a restrictive assumption is $d$ is moderate.
>
> While we focus on showing a lower bound for some scale of n and d (which at least implies an asymptotic lower bound), it is an interesting open question to show a lower bound when n is much smaller. Note that this scaling of n is only required to show that Assumption 1 is satisfied by the lower bound. If we were willing to discard this assumption, the lower bound holds without requiring $n \ge d 2^{d+3}$.
>
> > Requested Changes
>
> Thanks for the suggestions. We have updated the paper with the changes requested.

---

### Review · Reviewer_HJac · 2023-02-10

**Summary Of Contributions:**

The authors analyze the learning of pseudo-Boolean functions, i.e., decision functions from $\\{-1, 1\\}^d\rightarrow \mathbb{R}$. The latter are known to admit a polynomial representation, whose associated coefficients have been empirically shown to be sparse. The authors thus analyze the statistical properties of the minimizers of the empirical mean squared error, penalized with the $\ell_1$ norm of the coefficients. Under some curvature (Assumption 1(a)) and smoothness (Assumptions 2, 3) conditions, the authors show that for a suitably chosen regularization parameter any first order stationary point of the regularized objective is close to the ground truth one (Theorem 1). The authors then relax the curvature assumption into Assumption 1.(b), but consider the more demanding $\Delta$ first order stationary interpolators (Definition 3). The latter have similar guarantees in this relaxed setting (Theorem 2). A lower bound in expectation is also presented (Theorem 3), and experiments carried out in Section 5.

**Audience:**

Yes

**Claims And Evidence:**

No

**Requested Changes:**

See weaknesses above

**Strengths And Weaknesses:**

**Strengths**
- The paper is globally clear and well written
- The topic is of interest to the TMLR community


**Weaknesses**
- It is not entirely clear to me how the presented analysis (especially that of Theorem 1) differs from that of LASSO, or already known techniques. In particular, it seems that Assumption 1(a) somehow allows to shortcut the difficulty of considering models beyond the linear setting. Further, the authors do not present any other setting in which the assumption is satisfied. I feel it would be necessary to come up with other examples of interest to showcase that the presented analysis truly goes beyond the linear setting.
- I read the discussion in Appendix C but really do not appreciate the wording "for a sufficiently large $C_{0/1}$". If they can be set explicitly to 10, I feel it should be done. In the current version, it is not clear whether these constants are the same or not from one proposition to the other. More importantly, it suggests that we can take $\lambda$ in Theorem 1 as big as we wish and the result would still hold, which I assume is wrong. This inaccuracy is easy to clarify by expliciting the values.
- Regarding Theorem 2, could the authors provide some examples of algorithms satisfying Definition 3 with the required $\Delta$? Also, how come $\Delta$ does not play any role in the bound (13)? I feel constants should be handled more carefully here to highlight the dependencies. Also, (13) does not depend on the confidence level $\delta$ nor the variance $\sigma$, is it expected? The $1/2$ seems not necessary if authors use $\lesssim$. I encourage the authors to proof read the statement of Theorem 2.
- The lower bound (Theorem 3) is in expectation. Also, it could be beneficial if the authors could provide some intuition about its construction.
- Have the authors tried to empirically test the lower bound?
- In the Appendix, the authors refer many times to some event $\cal{E}_1$, which I could not find defined/introduced. Please clarify this.


**Minor comments**
- use \citep to put references between parenthesis when it is not part of the sentence
- p.3 paragraph "Problem statement": instead of assuming the existence of $\cal{F}$, let $\cal{F}$ and assume the existence of some $\theta^* \in \mathbb{R}^m$ such that labels are generated...
- use "with probability greater / at least $1-\delta$" instead of "probability $\ge$"
- $f_\theta$ instead of $g_\theta$ in Lemma 1?
- $\mu^2$ and not $\mu$ in eq. (19)
- $2L^2 2^d \\|\hat{\theta} - \theta^*\\|_2^2$ in eq. (20)
- Hoeffding's inequality if for bounded r.v., what is used for eq. (42) is more the concentration of subGaussian r.v.
- Between eq. (73) and (74), $\lambda / \sqrt{2^d}$ is missing twice in the definition of $R(\theta)$
- Proposition 1: **in** the Cartesian product

---

> ### Author Response · Authors · 2023-02-24
> **Fundamental differences with LASSO and notes on the bounds**
>
> We thank the reviewer for their time, valuable insights, and encouraging notes. We have detailed our response to their comments and updated the draft accordingly:
>
> > It is not entirely clear to me how the presented analysis (especially that of Theorem 1) differs from that of LASSO, or already known techniques. In particular, it seems that Assumption 1(a) somehow allows to shortcut the difficulty of considering models beyond the linear setting. Further, the authors do not present any other setting in which the assumption is satisfied. I feel it would be necessary to come up with other examples of interest to showcase that the presented analysis truly goes beyond the linear setting.
>
> If the learned function is parameterized by its pseudo-boolean representation, then the existing analyses for LASSO can be used to show generalization bounds. However, the key difference in our analysis is guaranteeing that sparsity in the “spectral domain” is ensured, not sparsity in the parameter space (which is what LASSO guarantees). Many of these analyses use the fact that the LASSO regularizer itself is a convex function of the parameter $\theta$. However, this crucial property is not satisfied by the SP regularizer, which is a non-convex function of $\theta$. Thus, even though it may be natural to expect that sparsity in the spectral domain can be ensured by the nature of the SP regularizer, it is not at all clear that "stationary points" of the objective will satisfy this property due to this non-convexity, even under the RSI or QG conditions. This is a major fundamental difference with LASSO. Another way to notice that new techniques are required follows from the required lower bound $n$ $\geq$ $n_0$ on the dataset size in Theorems 1 and 2. This minimum dataset size requirement does not appear in previous works, such as Loh et al. (2014) for the analysis of LASSO.
>
>
> > I read the discussion in Appendix C but really do not appreciate the wording "for a sufficiently large $C_{0/1}$". If they can be set explicitly to 10, I feel it should be done. In the current version, it is not clear whether these constants are the same or not from one proposition to the other. More importantly, it suggests that we can take in Theorem 1 as big as we wish and the result would still hold, which I assume is wrong. This inaccuracy is easy to clarify by expliciting the values.
>
> We shall clarify the constants in Theorem 1 and 2. They can indeed be set explicitly to be at most 10.
>
>
> > Regarding Theorem 2, could the authors provide some examples of algorithms satisfying Definition 3 with the required? Also, how come $\Delta$ does not play any role in the bound (13)? I feel constants should be handled more carefully here to highlight the dependencies. Also, (13) does not depend on the confidence level $\delta$, nor the variance $\sigma$, is it expected? The 1/2 seems not necessary if authors use $\lesssim$. I encourage the authors to proofread the statement of Theorem 2.
>
> The current statement in eq(13) is slightly incorrect and has been changed to match that in eq (8), $\frac{\sigma L}{C_{n,\delta}^*} \sqrt{\frac{k(d + \log(1/\delta))}{n}}$. The last step of the proof incorrectly substituted the value of $\lambda$ to result in the previous version of eq. (13). The new expression in eq. (13) still does not depend on $\Delta$ which is interesting. This means that when the training error is smaller than a constant, the stationary points of the algorithm generalize well. The interpretation of this result is that there is a basin where all stationary points lying within that local basin generalize well, and all other stationary points lying outside the basin need not. While we do not produce an algorithm for finding such stationary points, the motivation for this assumption is precisely from the behavior observed in practice - overparameterized models learn to achieve nearly $0$ training error on the data. The interpretation of the result is that these approaches do not suffer from poor generalization as predicted by the standard generalization bounds. Under certain conditions, it is also possible to show that regular gradient descent can approach such stationary points. If the training loss is smooth and convex and the function class $f_\theta$ is bounded for example (the overall regularized loss is still non-convex), since $\lambda$ are small, it is possible to show that the stationary points of the regularized loss and the unregularized loss are not far apart (in that each stationary point of the regularized loss is near a stationary point of the unregularized loss). Therefore, running gradient descent on the regularized loss falls close to a stationary point of the unregularized loss. We are guaranteed that these points have small training errors (by the convexity of the training loss) and the guarantees in the paper carry through.

---

> ### Author Response · Authors · 2023-02-24
> **Lower bounds, intuitions, and minor comments**
>
> > The lower bound (Theorem 3) is in expectation. Also, it could be beneficial if the authors could provide some intuition about its construction.
>
> We believe the current techniques can be extended to a high probability version, which we leave as future work.
>
> An intuition for the lower bound is added to the paper: The key idea is that the most challenging instances of the problem come from when the function class is parameterized by its coefficients in the spectral domain. In this case, the true function f* may be any function with a k-sparse spectral representation, and the lower bound follows by showing that the log-covering number is the right notion of the complexity of the problem. The analysis resembles lower bounds for sparse L1 regression lifted to $2^d$ dimensions. We have included this in the paper.
>
> > Have the authors tried to empirically test the lower bound?
>
> This is a lower bound on the statistical error of the best algorithm against its worst-case instance. Empirically testing the lower bound involves identifying the best algorithm, which is unclear how to do. In Theorems 1 and 2, we showed that our algorithm approximately matches the lower bound in Theorem 3. The empirical evaluation of our algorithm has been shown in Figures 1 and 3.
>
> > In the Appendix, the authors refer many times to some event $\mathcal{E}_1$, which I could not find defined/introduced. Please clarify this.
>
> This has been addressed in the paper.
>
> > Minor comments
>
> Thanks for the suggestions. We have updated the draft.

---

### Review · Reviewer_QbBF · 2023-02-10

**Summary Of Contributions:**

This paper show that $\ell^1$-regularizing the so-called spectral representation helps to to learn (pseudo)-Boolean functions.
More precisely, the authors provide some theoretical results indicating that under the quadratic growth, this allows to find an (approximate) stationary point with $\sqrt{kd/n}$ error with high probability. This is also shown empirically on examples.

**Audience:**

Yes

**Claims And Evidence:**

Yes

**Requested Changes:**

- Please use \citep and \citet properly.
- The condition on $\lambda$ (big enough) is sound, but don't you risk to have for instance a solution reduced to 0 in the linear case if you do not control how big $\lambda$ (or $C_0$) grows?
- I would like to see a discussion on a the computational cost of MSE+SP w.r.t. to using for instance the Lasso.
- What do you mean by "empirical lower bound" in the description of Fig 2? Do you meant the minimal value achieved in the expriments? It is not perfectly clear for me how to use this information to derive a "true" lower bound.
- I don't undertand "specific path of regularization parameter \lambda". In my mind, the regularization path is the map $\lambda \mapsto argmin (OBJ)$ (where argmin should be better defined). Do you mean a particular trajectory of this potential multivalued map?
- In your source code, you should check if the directory are created, I had to manually use mkdir to be able to run your experiments.
- Assumption 2 should read $-\mu I \preceq \nabla^2 f_{\theta(x)} \preceq \mu I$.


**Strengths And Weaknesses:**


I believe the paper is technically sound and, despite the strong claim of the title with respect to the actual results, provide an interesting perspective on learning boolean functions under a sparse hypothesis.

**Strenghts**

- The proofs seem correct.
- The theoretical analysis under QG is interesting.

**Weaknesses**

- The paper does not a very good job at comparing the guarantees obtained for spectral regularizatin w.r.t for instance the Lasso.
- The authors recognize themselves that the bounds might not be really practical, but I would say that this is not the first paper in this type of work having this shortcoming...

---

> ### Author Response · Authors · 2023-02-24
> **Comparisons with LASSO**
>
> > The paper does not a very good job at comparing the guarantees obtained for spectral regularizatin w.r.t for instance the Lasso.
>
> Remark 5 in the paper, which is stated for the global minimizer of the regularized loss, can also be used to get a guarantee for LASSO, which we avoided stating in the interest of space. The generalization guarantee of LASSO is also $\widetilde{O} (\sqrt{kd/n})$. But LASSO cannot be implemented for learning sparse pseudo-boolean functions since it requires writing down the function as parameterized by $2^d$ coefficients, its spectral representation. By extending LASSO to account for a restricted function class $f_\theta$ which can be stored and computed more efficiently than computations utilizing $2^d$ coefficients (such as if $f_\theta$ represented a neural network), we arrive at the SP regularizer
>
>
> > The authors recognize themselves that the bounds might not be really practical, but I would say that this is not the first paper in this type of work having this shortcoming...
>
> Although the bounds may be improvable for special function classes, in the worst case, the guarantees are shown to be tight by the lower bound in Theorem 3. As we discuss in the subsection “Extension to the case of multiple local minima” in Section 4, these bounds can also be appropriately extended to the case where the function class satisfies symmetries (such as permutation invariance in neural networks) rendering the RSI and QG assumptions unusable.

---

> ### Author Response · Authors · 2023-02-24
> **Requested changes**
>
> > Please use \citep and \citet properly.
>
> This has been addressed in the paper.
>
> > The condition on $\lambda$ (big enough) is sound, but don't you risk to have for instance a solution reduced to 0 in the linear case if you do not control how big $\lambda$ (or $C_0$) grows?
>
> As we clarify in the response to the Reviewer 1, $C_0$ is an absolute constant <= 10 (and in the paper we choose it as $=3$). Therefore, the bound on $\lambda$, $4 C_0 \sigma \sqrt{(d + \log(1/\delta))/n}$ does not grow unbounded. In the cases where $\sigma \to \infty$, $d/n \to \infty$ or $\log(1/\delta)/n \to \infty$, outputting the $0$ solution is valid since the resulting generalization bounds we prove in Theorem 1 and 2 also go to $\infty$ with these parameter scalings. In these cases, the interpretation is that there are too few samples to have meaningful generalization to the required error probability, $\delta$.
>
> > I would like to see a discussion on a the computational cost of MSE+SP w.r.t. to using for instance the Lasso.
>
> Running LASSO naively has a computational cost scaling as $O(\textsf{poly} (2^d))$ since the underlying regression problem is of dimension $2^d$. Moreover, it is unclear how this can be reduced without some form of function approximation. The computational complexity of MSE + SP regularization was recently studied in this paper (Aghazadeh, Nature Comm. 2021). In the paper, we have included a discussion in Section 4.3 that shows another algorithm, based on sparse WHTs and a specific initialization which is observed to work in practice.
>
> > What do you mean by "empirical lower bound" in the description of Fig. 2? Do you meant the minimal value achieved in the expriments? It is not perfectly clear for me how to use this information to derive a "true" lower bound.
>
> The empirical lower bound in Fig. 2 is indeed a bound on $C^*_{n,\delta}$ obtained experimentally. Note that $C^*_{n,\delta}$ itself is computationally hard to estimate (it is similar to computing the exact Lipschitz constant of a function, which is not tractable) and so we resort to evaluating random perturbations of the function to bound $C^*_{n,\delta}$. $\hat{C}^*_{n,\delta}$ is likely to be more accurate as more and more random perturbations are evaluated.
>
> > I don't understand "specific path of regularization parameter $\lambda$". In my mind, the regularization path is the map (where argmin should be better defined). Do you mean a particular trajectory of this potential multivalued map?
>
> The terminology “specific trajectory of $\lambda$” must be clarified in the paper. Although the algorithm specifies a single value of $\lambda$, in practice, one can always run gradient descent on the regularized loss, with $\lambda$ being a function of time, as long as $\lambda(t)$ converges to a limit. This is what is referred to as “specific trajectory of $\lambda$. This has been changed in the paper to be more clear.
>
> > In your source code, you should check if the directory are created, I had to manually use mkdir to be able to run your experiments.
>
> Thank you for the information. The edits for the source code have been made.
>
> > Assumption 2 should read $- \mu I \preceq \nabla^2 f_{\theta(x)} \preceq \mu I$
>
> This typo as well as one in the statement of Theorem 2 has been fixed (see the response to Reviewer 1).

---

### Review · Reviewer_BvRF · 2023-02-15

**Summary Of Contributions:**

The paper studies the learning of pseudo-boolean functions (functions from $\{-1, 1\}^d$ to $\mathbb{R}$)/
Such functions can be written $f(x) = \sum_{\mathcal{S} \subset [d]} \alpha_{\mathcal{S}} \prod_{i \in \mathcal{S}} x_i$.
When learning such a function through empirical mean minimization, the paper proposes to $\ell_1$-regularize the spectral transform of the learned function.
This approach is motivated by the observation that the spectral transforms of several real-world pseudo-Boolean functions have a sparse structure.
Traditional sparsity approach may fail, as the number of spectral coefficients in the decomposition is $2^d$.

When the true empirical loss, $\sum_i (f_\theta(x^i) - f_{\theta^*}(x^i))^2$ satisfies a restricted secant condition (RSI, def 1), any critical point of the regularized objective is, w.h.p., close enough to the true parameter, with error bounds similar to that of the Lasso (Eq 8, 9) ($\sqrt{kd / n}$)
Without the RSI condition, an analysis under quadratic growth is provided, for "almost interpolating" critical points (and not just any critical point).

**Audience:**

Yes

**Claims And Evidence:**

No

**Requested Changes:**



## Experiments
- In figure 1, the generalization is considered a success if "the models generalize on unseen data with the coefficient of determination larger than R2 ≥ 0.45 on the test data points.". Why this value of 0.45? Can you directly plot the left-out error or the R2? rather than thresholding?
- The experiment on QG conditions seems flawed to me: why would a random sampling provide a good estimate of the direction in which the growth is minimal? If it were performed on a simple quadratic function, you'd need to sample in the direction of the eigenvector associated with the smallest eigenvalue; any isotropic sampling will vastly overestimate the smallest eigenvalue, and instead yield the average eigenvalue. Can the authors clarify?
- In the real world experiments, what does "Lasso" correspond to? What does "L1 norm penalty on the coefficients in the
polynomial representation" mean here; what are the polynomials?
How is $\lambda$ chosen for SP in this experiment? It seems that it is not tuned for other methods, so it should be clear that it's a vanilla choice here too.
- The labels of some plots are of low quality (Fig 7 ylabel and xlabel for example). matplotlib can be configured to use latex.

## Literature
The paper puts on the same level the Lasso (a statistical estimator) and FISTA (an optimization algorithm, suited amongst others to solving the Lasso minimization problem).
Similarly, suggesting that the Lasso objective "can be efficiently solved by gradient descent" is inexact, to say the least: 1) Gradient descent cannot be used, because of the non-differentiability of the L1 norm along the axes. 2) *sub*gradient descent could be used, but it performs very poorly and cannot be called "efficient", with a convergence rate in $1 / \sqrt{k}$.

## Typos
- In equation 19 $\mu$ should be $\mu^2$
- In equation 20 $\Vert \theta - \theta^* \Vert$ should be squared
- There are multiple useless parentheses that make the reading harder than it should, eg around the sqrt in 62, around the gradient in 48, and others.


## Formulations
- Some sentences/formulations seem somehow off, e.g. "ML models are being used" > ML models are used?
Running Grammarly on the paper may easily fix a lot of these.
- I couldn't understand "and which nonlinear function classes are often implicitly biased towards learning."
- positively correlated with the $\theta - \theta^*$
- each with an equal coefficients
- The authors use $(\nabla) R$ to denote the subgradient; using $\partial R$ (and without parenthesis) is more standard in the optimization literature. In proposition 1, this is not the subgradient, this is the subdifferential.
- Before eq 52, "Finally, using the convexity of": I see the dual representation of the L1 norm rather than its convexity.


**Strengths And Weaknesses:**

## Strength
- topic of interest for TMLR audience
- results on par with the existing literature on the Lasso (though a more in depth comparison of the constants would be useful); the dependency on the number of samples, dimension and true sparsity makes sense and is proven to be a lower bound.

## Weaknesses
- see requested changes for experiments below

---

> ### Author Response · Authors · 2023-02-25
> **Phase transitions, LASSO, and Formulations**
>
> > In figure 1, the generalization is considered a success if "the models generalize on unseen data with the coefficient of determination larger than R2 $\geq$ 0.45 on the test data points.". Why this value of 0.45? Can you directly plot the left-out error or the R2? rather than thresholding?
>
> We follow the definition of the phase transition curve in compressed sensing in plotting Fig. 1. For phase transition in compressed sensing, the probability of success is defined based on a threshold (see Figure 1 in Ref. [1] below). For better clarity, we showed the phase transition in our paper for the 0.45 threshold. We have also added the corresponding plot for the average of R^2 as suggested in Fig. 9. The conclusion remains the same: SP regularization improves the test accuracy in terms of $R^2$.
>
> [1] Donoho, David L., Arian Maleki, and Andrea Montanari. "The noise-sensitivity phase transition in compressed sensing." IEEE Transactions on Information Theory 57.10 (2011): 6920-6941.
>
> > The experiment on QG conditions seems flawed to me: why would a random sampling provide a good estimate of the direction in which the growth is minimal? If it were performed on a simple quadratic function, you'd need to sample in the direction of the eigenvector associated with the smallest eigenvalue; any isotropic sampling will vastly overestimate the smallest eigenvalue, and instead yield the average eigenvalue. Can the authors clarify?
>
> The random sampling scheme is not efficient but not flawed. In the case of simple quadratic case, there may be more clever algorithms for estimating the constant. However, none of them are exact, and rely on some degree of handwaving. In general, for neural networks and complex nonlinear functions the sampling scheme is much more challenging. Local second-order approximation / looking at Hessian of the function at a single point, $\theta^*$ no longer captures the behavior of $C^*_{n,\delta}$.
>
> > In the real world experiments, what does "Lasso" correspond to? What does the "L1 norm penalty on the coefficients in the polynomial representation" mean here; what are the polynomials? How is $\lambda$ chosen for SP in this experiment? It seems that it is not tuned for other methods, so it should be clear that it's a vanilla choice here too.
>
> Lasso corresponds to fitting a multi-linear polynomial (as defined in the second paragraph of Section 2) to the data with an L1 norm regularization on the coefficients $\alpha_i$'s. In the revised manuscript, we have clarified this in the experiments. SP regularization is not very sensitive to $\lambda$ ; we have clarified that $\lambda$ is only slightly tuned consistently across baselines and SP regularization.
>
> > The labels of some plots are of low quality (Fig 7 ylabel and xlabel for example). matplotlib can be configured to use latex.
>
> We have revised all figures for clarity.
>
> > The paper puts on the same level the Lasso (a statistical estimator) and FISTA ...
>
> We have revised that sentence and discussion and now we avoid discussing LASSO and FISTA in the same algorithmic categories.
>
> > Typos and Formulations
>
> We have thoroughly revised the manuscript for grammatical errors and typos.

---

### Decision · Action_Editors · 2023-03-22

**Recommendation:** Accept as is

**Comment:**

This paper studies the learnability of pseudo-boolean functions under a structural assumption that its Fourier transform is sparse. The resulting regularized empirical objective is shown to have tractable properties under mild assumptions, both theoretically and through experiments.
Reviewers agreed that this paper is technically sound and provides an appealing perspective on the benefits of spectral regularisation via sparsity-inducing norms. They also acknowledge that the authors have taken the reviewers feedback into account to produce a much improved version.

**Audience:**

Yes, the results of this paper should be of interest to a broad audience interested in learning theory .

**Claims And Evidence:**

Yes, the claims are supported both by theoretical analysis and by adequate empirical evaluation.